# Concentration of heavy metals in pasteurized and sterilized milk and health risk assessment across the globe: A systematic review

Zahra Alinezhad[1,2‡], Mohammad Hashemi[1,2‡], Seyedeh Belin Tavakoly Sany [ORCID][3,4]*

1 Department of Nutrition, Faculty of Medicine, Mashhad University of Medical Sciences, Mashhad, Iran,
2 Medical Toxicology Research Center, Mashhad University of Medical Sciences, Mashhad, Iran,
3 Department of Health, Safety, Environment Management, School of Health Mashhad University of Medical Sciences, Mashhad, Iran, 4 Social Determinants of Health Research Center, Mashhad University of Medical Sciences, Mashhad, Iran

‡ ZA and MH contributed equally as first authors on this work.
* belintavakoli332@gmail.com, tavakkolisanib@mums.ac.ir

## Abstract

### Objective

Although milk and dairy products are almost complete food, they can contain toxic heavy elements with potential hazards for consumers. This review aims to provide a comprehensive report on the occurrence, concentration, and health risks of selected heavy metals in pasteurized and sterilized milk recorded worldwide.

### Methods

The Preferred Reporting Items for Systematic Reviews and Meta-Analysis (PRISMA) was used to develop this systematic review. Databases included the Web of Knowledge, Scopus, Scientific Information Database, Google Scholar, and PubMed from inception until January 2023. Keywords related to the terms "Heavy metals", "Arsenic" and "Pasteurized and sterilized milk" and "Risk Assessment" were used. The potential health risks to human health from milk daily consumption were estimated using extracted data on heavy metals concentration based on metal estimated daily intake, target hazard quotient, and carcinogenic risk.

### Results

A total of 48 potentially relevant articles with data on 981 milk samples were included in the systematic review. Atomic Absorption Spectroscopy, atomic absorption spectroscopy, inductively coupled plasma-mass spectrometry, and inductively coupled plasma-optical emission spectrometry were the most common valid methods to measure heavy metals in milk samples. Following the initial evaluation, Cu, Cd, Zn, and Pb were the most contaminants, which exceeded the maximum permissible criteria in 94%, 67%, 62%, and 46% of the milk samples tested. Relying on target hazard quotient and carcinogenic risk results, milk consumers in 33(68.75%) and 7 (14.5%) studies were exposed to moderate to high

**Data Availability Statement:** All relevant data are within the manuscript and its Supporting Information files.

**Funding:** The author(s) received no specific funding for this work.

**Competing interests:** The authors have declared that no competing interests exist.

levels of carcinogenic and non-carcinogenic risk, respectively. The highest level of risk is due to the consumption of pasteurized and sterilized milk detected in Pakistan, Brazil, Egypt, Slovakia, and Turkey.

## Conclusion

The elevated levels of heavy metals in milk samples, especially Pb and Cd is a public health concern; therefore, maximum control and strict regulations must be adopted to decrease heavy metals contaminants in the dairy industry. Further studies are required to develop safe milk processing and handling methods for the decontamination of heavy metals in milk and its products.

## Introduction

Food safety is an important challenge to maintain people's health for disease control and prevent food contamination and causing food intolerance and food poisoning [1, 2]. As defined by the World Health Organization (WHO) and Food and Agriculture Organization (FAO), "food safety is a science-based discipline, process or action that prevents food from containing substances that could harm a person's health" [2]. Milk and dairy products are the main sources of macro- and micronutrients, such as essential fatty acids, amino acids and vitamins, which are nesessary for bone development, growth, and immune functions [3–5]. Consuming at least three of dairy products per day has a beneficial impact on energy and nutrient intakes as well as of vitamin D, magnesium, and calcium, compared with intakes of people who consumed fewer servings of dairy products per day [6]. Unsafe food containing harmful bacteria (Salmonella, Vibrio cholerae, enterohaemorrhagic Escherichia coli, and Campylobacter), viruses (Hepatitis A), parasites (tapeworms, Ascaris, Cryptosporidium, Giardia, and Entamoeba histolytica) or chemical substances (Persistent organic pollutants, heavy metals, mycotoxins, and radioactive nucleotides) causes more than 200 diseases such as diabetes, respiratory problems, hypertension, coronary heart disease, stroke, and colorectal and bladder cancer [7–9]. It also promotes the immune system, good bone health, and the prevention of dental caries [8, 10].

Milk provides essential nutrients and energy for proper growth and development, and its consumption goes from infancy to old age. It is valued not only for its nutrient contribution but also that of other bioactive compounds [1]. In adults of North America, milk consumption amounts to approximately 267 L/year [2]. In Peru, consumption per person is 87 L/year, well above the European average of 59.4 L/year, while Germany reports 53.7 L/year [3]. In last decade, milk consumption has increased by about 20% globally, and 48% of the total milk products were represented by Bovine milk. Milk and dairy products can contain residues of hazardous chemical or biological pollutants with potential hazards for consumers [11–13]. The risk of biological pollutants derives from cattle milking includs the exposure of udders to the environmental pollutants, storage, equipment, and dirty pipes [14, 15]. Chemical pollution of milk comes from use of illegal or legal veterinary products, forages and feed contaminated with natural toxin, application of agrochemicals, and the improper use of chemicals during processing, production, packaging, storage, and handeling, storage, and even pasteurization of milk [16, 17]. Based on the literature, the most reported contaminantes that treatens the safety of milk are pathogenic micoorganisms (14.57%), heavy metals (22.18%), antibiotics (22.18%),

pesticides (22.05%), and mycotoxins (9.97%). Therefore, heavy metals are stand out among contaminants [7–9].

Stable chemical pollutants such as heavy metals are the major food contaminants in the world. Heavy metals are elements with a high specific gravity or atomic number and are attributed to have a specific gravity of $> 5$–6 g/cm$^3$ or an atomic number of 63.5–200 g/mol [11, 18, 19]. Some heavy metals like Fe, Cu, Zn, and Mn, are essential elements for human health with critical role in metabolism and biochemical functions in the human body; but their consumption at concentrations higher than the sanitary recommendations imposes adverse effects on human health in terms of toxicology [19–21]. Other heavy metals, including Pb, Cd, Hg, and As, are non-essential and toxic elements and can cause adverse effect on human health even at low level [12, 22–24]. Heavy metals have been of interest due to such properties as toxicity, high dispersion, high thermal resistance, high level of bioaccumulation along the food chain, and their non-biodegradability [19, 20, 22, 25]. At high concentrations, these stable pollutants can have adverse effects, such as toxicity, mutagenesis, and carcinogenesis [12, 22–24]. The International Agency for Research on Cancer (IARC) has classified Cd, As, and Pb as carcinogens in groups 1 and 2, respectively [3]. The use of heavy metal-contaminated food is a major entry route of these contaminants into the body [3].

Although milk and dairy products are almost complete food, they can contain toxic heavy elements with potential hazards for consumers. It was evidenced that the presence of heavy metals in milk and dairy products was detected in different countries across the world. However, the level of heavy metals concentration is not differed and is constant depending on animal's feed, environmental condition, the production system, exposure routes, use of agrochemicals, breed of cattle, and stage of lactation [3, 18]. Several sources contaminate milk and dairy products with heavy metals that mainly originate from environmental pollution that contaminate water and animal feeding, anthropogenic activity, during the packaging and storage of milk products [11–13, 26]. Moreover, raw milk is prone to contamination with metals in the factory during different milking processes, including equipment contamination. These metals are somewhat reduced by such procedures as fat separation during pasteurization and sterilization processes, but they are not removed completely [12, 17]. Although, pasteurization has remained as an efficient treatment method to eliminate contaminants, many infectious diseases, associated with pasteurized milk occur in raw milk with an exaggerated contamination of chemical and biological compound [27, 28]. The main problem in pasteurization is that this method only efficient for reducing the levle of most biological and non-chemical contaminants [29].

Based on the literature, very few alternative methods in the of chemical contaminants in cow's milk, cause to relevent analysis to ensure their sufficient quality in the milk products [27]. In recent decades several studies reported the occurrence of heavy metals contamination in milk even in pasteurized milk products [11–13]. However, most of these studies have not provided comprehensive reports and trends on the distribution of heavy metals contamination and their health risk in pasteurized and sterilized milk across the world. Public health need to be well-informed about levels of heavy metal contamination and possible cancer or non-cancer risks presented in milk and dairy products. Therefore, this systematic review aims to provide a comprehensive report on the occurrence and concentration, of selected heavy metals including aluminum (Al), copper (Cu), iron (Fe), Zinc (Zn), Cobalt (Co), and nickel (Ni) as well as toxic heavy metals involving Arsenic (As), cadmium (Cd), mercury (Hg), lead (Pb), and Chromium (Cr) in the pasteurized and sterilized milk recorded across the world. In addition, the potential carcinogenic and non-carcinogenic risks to human health from milk daily consumption were estimated using extracted data on heavy metals concentration.

The main target of this review to provide a comprehensive report on the occurrence and concentration of selected heavy metals in the pasteurized and sterilized milk recorded across the world. We did not inclcuded the raw milk in this review because several reviews were conducted on raw milk. But levels of heavy metal contamination, their possible health risks, and temporal trend in the pasteurized and sterilized milk is not clear. In addition, the potential carcinogenic and non-carcinogenic risks to human health from milk daily consumption were estimated using extracted data on heavy metals concentration.

## 2. Materials and methods

### 2.1. Study design

The Preferred Reporting Items for Systematic Reviews and Meta-Analysis (PRISMA) was used to develop this systematic review [30] (**S1 Checklist**). The main outcome of interest for the review the concentrations of heavy metals and trace elements in the pasteurized and sterilized milk reported across the world and potential health risk to human health (carcinogenic and non-carcinogenic effect) and trace elements estimated daily intake of milk were also extracted from different studies.

### 2.2. Search strategy

In this study, databases were searched, including the Web of Knowledge, Scopus, Scientific Information Database (SID), Google scholar and PubMed from inception until January 2023. The search was independently conducted for each database, and references of studies were cross-checked for identification of literature that related to concentration of heavy metals in Pasteurized and sterilized milk and health risk assessment across the globe. The reference list of all eligible studies was also hand-searched to find other relevant studies that may have been ignored by the search process. Different keywords related to the term "heavy metals", "Arsenic" and "Pasteurized and sterilized milk" and "Risk Assessment" were searched.

### 2.3. Inclusion and exclusion criteria

In this review, studies were included if they carried out in world, published in the Persian and English language, reported concentration of heavy metals in Pasteurized and sterilized milk and health risk assessment across the globe, and focused on subjects such as the concentration of heavy metals and health risk assessment of heavy metals in world. The study selection for inclusion eligibility was conducted by scanning the titles, abstracts, and full texts of retrieved articles. All review studies and duplicate research were excluded.

### 2.4. Data extraction and quality assessment

In this study, a set of basic data were extracted as follow: the publication year/author, study objective, study area, period of sampling, sample size type(s) of samples, quality control/quality assurance, analytical technique, number of heavy metals and heavy metals concentration, and potential pollution sources. The quality of the selected studies was examined via Ofori and Cobbina adapted the Newcastle–Ottawa Scale (NOS). This scale includes six items that were provided with either a "no" or "yes" response to make a general score for each study (Table 2).

### 2.5. Risk assessment analysis

In this review health risk assessment was conducted based on Metal Estimated Daily Intake (EDI), Target hazard quotient (THQ), non-carcinogenic risk, and carcinogenic risk, which was calculated based on the mean levels of heavy metals that reviwed in the current review (Table 1).

**Table 1. Health risk assessment indices and variables based on EPA model [31].**

| Cutoff point | Variables | Formula | Risk Indices |
|---|---|---|---|
| Estimated Daily Intake (EDI)(mg/kg/day) | $EDI = \frac{(C \times IR \times EF \times ED)}{(BW \times AT)}$ | C (mg/kg), IR (gr/kg/day), EF equal 365 days per years, ED is 70 and 6 years for adult and child; BW (Kg); AT (day) | It is not exceeded provisional tolerable daily intake (PTDI) of heavy metals. |
| Target Hazard Quotient (THQ) Hazard Index (HI) | $THQ = EDI/RFD$ $HI = THQ_1 + THQ_2 + \ldots$ | RFD (mg/kg/day) is equal to As = 0.0003, Cd = 0.001, Al = 0.7, Co = 0.0004, Cu = 0.04, Fe = 0.7, Hg = 0.0004, Zn = 0.3, Cr = 0.003, Ni = 0.02, Pb = 0.0035, Ba = 0.07 | THQ and HI < 1 indicating no non-carcinogenic risk from exposure to toxic elements/THQ and HI > 1 indicating possible non-carcinogenic risk |
| Cancer Risk (CR) | $CR = EDI \times OSF$ | OSF (mg/kg/day-1) is equal: As = 1.5, Pb = 0.0085, Cd = 0.38, Hg = 0.0003 | $CR < 10^{-4}$ indicating -carcinogenic risk from exposure to toxic elements/ $CR > 10^{-4}$ indicating possible carcinogenic risk |

C: Concentration; IR: Ingestion Rate; EF: Exposure frequency; ED: Exposure duration; BW; AT: Averaging time; RFD: Oral Reference Dose; OSF: Oral Slope Factor; PTDI: The value is not exceeded provisional tolerable daily intake of heavy metals

## 3. Results and discussion

### 3.1. Search outcome and quality assessment

The inclusion and exclusion criteria were implemented to select eligible studies (**Fig 1**). A total of 1349 published articles were selected for evaluation from which 48 studies with data on 981

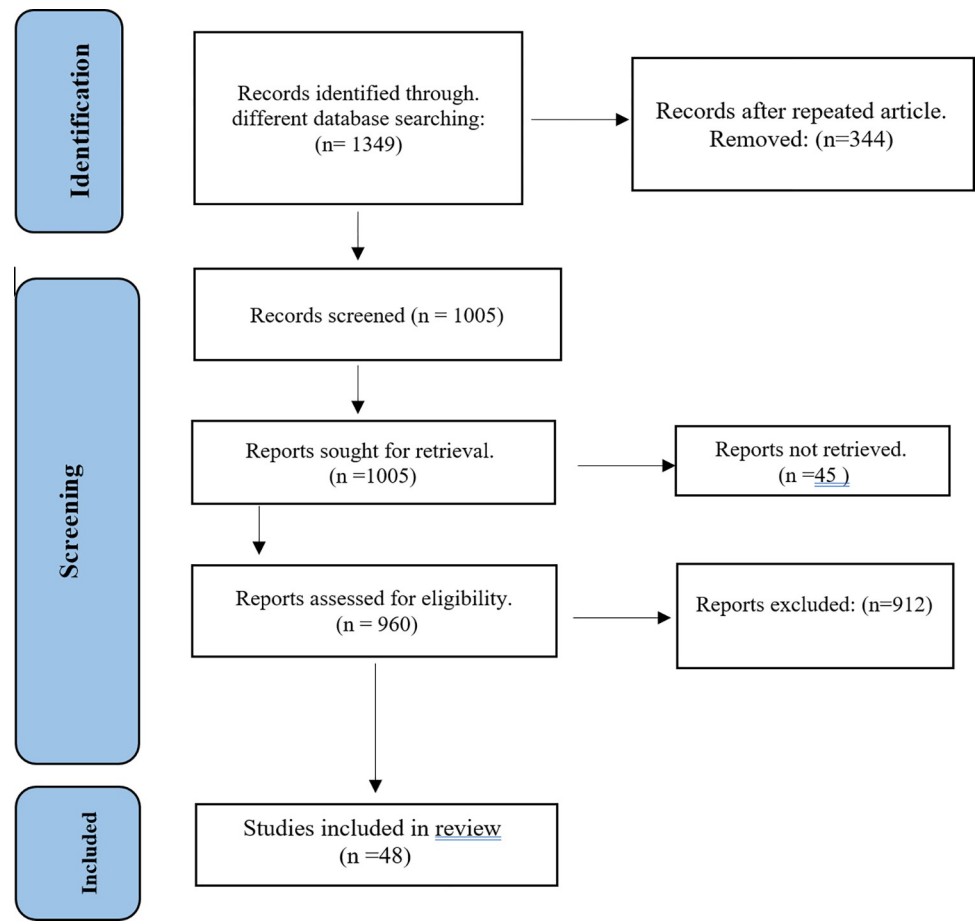

**Fig 1. Fellow diagram of the studies selection process following the PRIZMA.**

samples were met inclusion criteria and included for the final report. All information was extracted based on location, Sample size, apparatus, heavy meals concentration and outcomes. For the quality assessment of eligible articles, about 23% of the studies showed that appropriate quality and ranged from 5 to 6 out of 6 [5, 11, 21, 22, 32–38], All these studies used a standard protocol for sampling, period of sampling, quality control, all objectives achieved and mean heavy metals levels. The quality of 29(60.4%) studies was moderate with an average score of 4 [3, 12, 18, 20, 23, 24, 39–61], and about 8(16.6%) of the articles showed low quality (from 2 to 3) [4, 25, 62–67] and their major reasons were: a lack of standard protocol for sampling and non-indication of heavy metals sources (**S1 Table**).

## 3.2. Study samples characteristics

A total of 48 potentially relevant articles with data on 981 pasteurized and sterilized milk were included in the systematic review. All data were collected from 17 different countries including Iran (13:27%) [3, 5, 12, 20, 23, 24, 32, 36, 41, 43, 48, 50, 62], Pakistan (6:12.5%) [4, 11, 42, 55, 58, 63], Egypt(5:10%) [18, 40, 45, 53, 56], and Brazil (4:8%) [22, 46, 54, 64] represented the highest number of studies, and the sample size was varied from 5 to 90. Based on results, the first study was conducted in 1993, and 70% of the studies were done in the past five years.

A number of techniques are used to quantified heavy metals in the aqueous phase which related to the desired level of detection and the specific analyte. Our finding showed that from 48 included articles, 15 techniques were used to measure heavy metals in pasteurized and sterilized milk include; Atomic Absorption Spectroscopy (AAS)(15:31.3%) [11, 18, 33, 40–43, 50, 51, 53, 55–57, 61, 67], Microwave Plasma–Atomic Emission Spectrophotometer(MP-AES) (1:2.1%) (21),Metrohm 797 VA Computrace(1:2.1%), Inductively Coupled Plasma Atomic Emission Spectroscopy (ICP-AES)(2:4.15%) [25, 32], Graphite Furnace Atomic Absorption Spectroscopy(GFAAS) (2:4.15%) [5, 35], Flame Atomic Absorption Spectrophotometer (FAAS)(3:6.3%) [39, 46, 64], Inductively Coupled Plasma-Mass Spectrometry (ICP-MS) (7:15%) [34, 38, 44, 45, 49, 52, 65], Inductively Coupled Plasma-Optical Emission Spectrometry (ICP-OES)(4:8.5%) [4, 22, 37, 62], polarography apparatus(1:2.1%) [24], voltammeter (1:2.1%) [20], Laser Induced Breakdown Spectroscopy (LIBS)(1:2.1%) [48], Inductively Coupled Plasma Sector Field Mass Spectrometry (ICP-SFMS)(1:2.1%) [36], Anodic Stripping Voltammetry (ASV)(1:2.1%) [54], Electrothermal Atomic Absorption Spectrophotometer (ET-AAS)(2:4.15%) [58, 66] and Metrohm 746 VA trace(1:2.1%) [59], and 5 studies(5:10.5) [3, 23, 47, 60, 63] did not report the Apparatus.

Atomic Absorption Spectroscopy (AAS)(15:31%), ICP-MS, and ICP-OES were the most common valid method used to measure heavy metals in milk in the globe, while other tools were rarely implemented. The atomic absorption spectrophotometer and FAAS are most commonly used to quantify the heavy metals in the aqueous phase such as milk by the developing countries due to its low cost, easy operating process, and rapid detection. But, ICP-MS or ICP-OES have been impelemented for more sensitive measurment of trace elements. However, this method need high costs to run; therefore, it most commonly used in developed countries.

## 3.3. Heavy metal concentrations in pasteurized and sterilized milk

**3.3.1. Pb.** Lead is not only non-essential for living organisms but also has toxic properties, and its accumulation in the human body causes serious consequences for humam health [68, 69]. Infant and children are vulnerable groups to Pb poisoning because childeren absorb four to five times as much ingested Pb as adults from pathways [70, 71]. High concentraten of exposure Pb damages the central nervous system and brain, causing convulsions, and coma, intellectual disability, and even death [72, 73].

The standard Pb limit in milk is 0.02 mg/l set by Codex Alimentarius Commission and FAO/WHO standard, and EU [74, 75]. In this systematic review, the mean Pb levels in pasteurized and sterilized milk samples are analyzed in 41 studies (85.5% of total studies) from 13 countries Pakistan [4, 11, 42, 55, 58], Libya [21, 35], Iran [3, 5, 12, 20, 23, 24, 32, 36, 41, 43, 48, 50, 62], Egypt [18, 40, 45, 53, 56], Indonesia [33], Brazil [22, 46, 54, 64], China [37, 44, 49], Serbia [47], North Korea [52], Romania [25, 38], Turkey [59], Spain [65, 66], and India [60] from 1999 to 2021 (Table 2).

The average concentration of Pb in pasteurized and sterilized milk samples across globe ranged between $11 \times 10^{-5}$ [25] and 3.5 mg/ml [22]. Lead was the most common heavy metal in pasteurized and sterilized milk samples found above permissible limits (0.02 mg/l) [74, 75] in 19[46.34%] studies were conducted in Libya [21], Iran [12, 43, 62], Egypt [18, 40, 45, 53], Pakistan [4, 42, 58], Brazil [22, 46, 54, 64], Serbia [47], Turkey [59], China [37] and Spain [66].

The highest Level of Pb levels in milk samples were detected in Brazil(3.5±3.3 mg/L) [22] and Egypt(3.5±1.5 mg/L) [45]. In these studies, the high Pb concentrations were observed in milk samples obtained from farm located in contaminated areas [22, 45]. Similarly, a high Pb level (2.9±0.05) was determined in Turkey [59]. This mainly results from atmospheric pollution, excessive use of fertilizers and pesticides, contamination during production and packaging processes [59]. Likewise, Pb generally comes from NPK fertilizers and calcium ammonium nitrate fertilizers that contains up to 12 mg/kg and 24.6 mg/kg of Pb.

In Egypt, Eman Abdelfatah et al., (2019) observed a high Pb content (1.9±0.2 mg/L) in milk samples which exposed to Pb through contaminated water or wastewater for agricultural purposes [40]. This contaminated water may be directly used by livestock, especially dairy animals, through drinking water or by bioaccumulation in soil, fodder, and vegetables [40]. Furthermore, high levels of Pb were obtained from sterilized milk samples collected from Brazil(0.7±1.2 mg/L) [46] and Egypt (0.7±0.04 mg/L) [53]. These studies reported that industrial activities, using contaminated feed, and the use of Pb-containing conserved cans are main sources of Pb [46, 53].

In Egypt, Mostafa et al. (2018) presented evidence that a high Pb concentration (0.4±0.1 mg/L) measured in pasteurized milk might be due to environmental contamination with this metal [18]. Suturovic´ et al. (2014) measured an average Pb concentration (0.09±0.007 mg/L) in Serbia, where the maximum standard Pb concentration in milk is 0.1 mg/kg. Their determined Pb concentrations in two pasteurized milk samples were close to the maximum permissible limit [47].

A elevated level of Pb in pasteurized milk samples was also reported in Spain(0.079±0.07 mg/L), which might result from contamination during the pasteurization process in the factory or during packaging [66]. A higher elevated Pb concentration was also recorded in three studies from Libya and Pakistan [4, 21, 58]. They reported that high concentration of Pb in all milk samples mainly results from environmental pollution in these areas such as atmospheric deposition, waste disposal, vehicle smoke, and urban wastewater [4, 21, 58].

Two studies were conducted in Iran reported that Pb concentration in 33% of milk samples (0.02±0.02 mg/L) were higher than permission level of WHO/FAO in all milk samples because all milk samples were collected from animal farms close to a lead-zinc smelting plant, mining waste storage, thermal power plants, and roads [12, 62].

The results of review highlighted that pasteurized and sterilized milk can a common source of Pb poisoning. The level of Pb in most of samples (70% to 100%) from developing countires (Iran, Egypt, Pakistan, Serbia, and Turkey) were beyond standard limit, less Pb contamination was found in milk samples from developed countries. This result mainly due to stricter implementation of regulations, accurate detection techniques, and controlled industrialization in developed countries compered with developing countries.

**Table 2. Heavy metals levels in pasteurized and sterilized milk reported in research articles published since 1993–2021.**

| Code | location | N | Apparatus | Heavy metals levels (mg/L) | | | | | | | | | | |
|---|---|---|---|---|---|---|---|---|---|---|---|---|---|---|
| | | | | Fe | Al | Ni | Cu | Pb | Zn | Hg | Cd | Co | Cr | As |
| 1 | Pakistan | 10 | AAS | – | – | – | – | 0.006 ±0.0002 | – | – | 0.002 ±0.0001 | – | – | – |
| 2 | Libya | 11 | MP-AES | 1.89±162 | – | 0.07 ±0.01 | **0.1±0.03** | **0.05 ±0.001** | 2.01 ±0.19 | – | ND | ND | 0.04 ±0.001 | – |
| 3 | Iran | 60 | Metrohm 797 | – | – | – | **0.9± 0.6** | **0.02±0.02** | **4.3± 1.65** | – | **0.02± 0.04** | – | – | – |
| 4* | Iran | 25 | ICP-AES | – | – | – | – | 0.008 ±0.005 | – | – | – | – | – | – |
| 5 | Iran | 18 | GFAAS | – | – | – | – | 0.01±0.003 | – | – | – | – | – | – |
| 6* | Iraqi | - | FAAS | 2.009 | – | – | **0.121** | - | **12.815** | – | **0.4± 0.13** | – | 0.151 | – |
| 7 | Egypt | 20 | AAS | – | – | – | **0.6 ±0.081** | **1.9±0.22** | – | – | – | – | – | – |
| 8 | Iran | - | – | – | – | – | – | 0.003 ±0.0006 | – | – | 0.0004 ±0.0004 | – | – | – |
| 9* | Egypt | 36 | AAS | – | – | 0.07 ±0.01 | **0.2 ±0.082** | **0.4±0.1** | 1.4 ±0.424 | – | **0.003 ±0.003** | – | 0.07 ±0.029 | – |
| 10 | Indonesia | 30 | AAS | – | – | – | – | 0.004± 0.002 | – | – | – | – | – | – |
| 11* | Turkey | 40 | ICP-MS | **1.9± 5.70** | 0.06 ±0.02 | 0.009 ±0.008 | 0.004 ±0.002 | – | 0.8 ±0.331 | – | | – | 0.01 ±0.02 | **0.0049 ±0.002** |
| 12* | Pakistan | 15 | ICP-OES | 1.98 | **3.35** | – | **0.49** | **0.051** | 2.69 | – | **0.004** | – | 0.09 | ND |
| 13* | Iran | 10 | AAS | 3.1±0.005 | – | – | **0.2 ±0.001** | 0.01 ±5×10−5 | – | – | **0.0098 ±0.00056** | – | – | **0.007 ±0.0007** |
| 14 | Libya | - | GFAAS | – | – | – | – | 0.01±0.006 | – | – | **0.09 ± 0.11** | – | – | – |
| 15 | Iran | 26 | PAV | – | – | – | **0.2 ±0.183** | 0.04±0.03 | 0.9 ±0.81 | – | **0.049±0.03** | – | – | – |
| 16 | Iran | - | | – | – | – | **0.3 ±0.014** | 0.009 ±0.0002 | 0.4 ±0.021 | – | **0.001 ±0.0001** | – | – | – |
| 17 | Pakistan | 15 | AAS | 1.6±0.051 | – | **0.14 ±0.041** | **0.02 ±0.021** | **0.19±0.13** | **6.8 ±2.06** | – | **0.04±0.02** | 0.05±0.04 | – | – |
| 18 | Iran | 20 | AAS | – | – | – | – | **0.04±10⁻⁶** | – | – | **0.006±10⁻⁶** | – | – | – |
| 19 | Brazil | 11 | ICP-OES | 1.3 ±.4.121 | – | **0.15 ±0.31** | **0.2 ±0.151** | 3.5±3.31 | **4.6 ±3.81** | 6.8 ±3.22 | – | 1.05±0.84 | – | – |
| 20 | China | 9 | ICP-MS | – | 0.17 ±0.09 | 0.05 ±0.08 | **0.04 ±0.01** | 0.006 ±0.002 | **2.8 ±0.44** | – | 0.0019 ±0.001 | 0.0038 ±0.0007 | 0.01 ±0.003 | – |
| 21* | Egypt | 90 | ICP-MS | – | 1.4 ±0.13 | – | **1.9±0.73** | 3.5±1.52 | 7.7 ±2.92 | – | **0.29±0.12** | – | – | – |
| 22* | Brazil | 9 | FAAS | 1±8.00 | – | – | – | **0.7±1.22** | 3±7 | – | – | – | **0.6±0.12** | – |
| 23 | Iran | 36 | ICP-OES | – | – | – | 0.008 ±0.005 | **0.02±0.02** | 0.09 ±0.09 | – | **0.005 ±0.009** | – | – | – |
| 24* | Serbia | 5 | – | – | – | – | **0.1 ±0.0009** | **0.09 ±0.007** | – | – | **0.004 ±0.0005** | – | – | – |
| 25* | Iran | - | LIBS | 8.92 | – | – | **10.6± 0.32** | 0.0008 | **28.1 ±8.32** | – | **0.0006** | – | – | **0.0008** |
| 26* | China | 8 | ICP-MS | – | – | – | – | 0.01 ±0.0008 | – | – | **0.003 ±0.0003** | – | – | – |
| 27* | Iran | - | AAS | – | – | – | **0.6±0.14** | 0.003 ±0.0019 | **5.06 ±0.39** | – | 0.00029 ±9×10⁻⁵ | – | – | – |
| 28 | Iran | 15 | ICP-SFMS | – | 0.13 ±0.08 | – | – | 0.01±0.007 | – | 0.02 ±0.009 | **0.0039 ±0.002** | – | – | **0.002 ±0.001** |
| 29 | Iran | 50 | – | – | – | – | **0.3 ±0.112** | 0.009± 0.001 | 0.4 ±0.12 | – | 0.001± 0.00049 | – | – | – |

*(Continued)*

**Table 2.** (Continued)

| Code | location | N | Apparatus | Heavy metals levels (mg/L) | | | | | | | | | | |
|------|----------|---|-----------|------|------|------|------|------|------|------|------|------|------|------|
| | | | | Fe | Al | Ni | Cu | Pb | Zn | Hg | Cd | Co | Cr | As |
| 30 | Slovakia | - | AAS | 1.9±1.41 | – | 1.01± 0.32 | 0.9 ±0.214 | – | 11.3 ±1.95 | – | 0.3± 0.21 | – | – | – |
| 31* | South Korea | 64 | ICP-MS | – | – | 0.1± 0.0002 | 0.3± 0.0005 | 0.01 ±3.8×10⁻⁵ | 4.7± 0.006 | – | 0.002 ±2.3×10⁻⁵ | 0.005 ±1.5×10⁻⁵ | 0.3 ±0.0004 | 0.0019 ±6.8×10⁻⁵ |
| 32 | Egypt | 10 | AAS | – | 1.8 ±0.07 | – | – | 0.7±0.04 | – | – | 0.3±0.01 | – | – | – |
| 33 | Romania | 8 | ICP-MS | – | – | 0.001 ±0.0005 | 0.2±0.08 | 0.01±0.008 | 3.2 ±1.04 | – | 0.01±0.005 | 0.001 ±0.0001 | 0.06 ±0.05 | – |
| 34* | Brazil | – | ASV | – | – | – | – | 0.2± 0.09 | 13.08± 0.4 | – | 0.07± 0.005 | – | – | – |
| 35 | Pakistan | – | AAS | 0.2 ± 0.11 | – | 0.006± 0.001 | 0.4 ± 0.5 | 0.0006± 0.001 | 1.9± 1.71 | – | 0.03± 0.02 | – | 0.0001 ±0.001 | – |
| 36* | Romania | – | ICP-AES | 11.82 | 4.16 | 0.05 | 0.131 | 11×10⁻⁵ | 1.52 | – | 5×10⁻⁶ | – | 0.1 | – |
| 37* | Egypt | 60 | AAS | 0.6±0.41 | – | 0.03 ±0.02 | 0.1±0.21 | 0.02±0.02 | 3.1 ±0.63 | – | 0.02±0.02 | 0.03±0.02 | 0.03 ±0.021 | – |
| 38* | Poland | – | AAS | 4.1± 0.22 | – | – | – | – | 3.79± 0.06 | – | – | – | – | – |
| 39 | Pakistan | 60 | ET-AAS | – | 1.8 ±0.13 | 0.2 ±0.012 | – | 0.05 ±0.004 | – | – | 0.05±0.002 | – | – | – |
| 40* | China | 32 | ICP-OES | 2.4±0.59 | – | – | 0.3±0.05 | 0.03 ±0.008 | 3.6 ±1.66 | ND | 0.004 ±0.003 | – | 0.1 ±0.053 | ND |
| 41* | Pakistan | – | – | – | – | – | 0.2 ±0.004 | – | – | – | – | – | ND | – |
| 42 | Brazil | 54 | FAAS | 1.05±0.89 | – | 0.07 ±0.07 | 1.7±0.82 | 0.2±0.12 | 4.59 ±1.31 | – | – | – | 0.079 ±0.052 | – |
| 43* | Turkey | – | Metrohm 797 | – | – | – | 0.09 ±0.0001 | 2.9± 0.05 | – | – | 0.6±0.0001 | – | – | – |
| 44 | Spain | – | ICP-MS | 0.18±0.01 | 0.03 ±0.004 | 0.01 ±0.001 | 0.02 ±0.003 | 0.002 ±0.0003 | 3.1± 0.24 | ND | 0.0005 ±0.00019 | – | 0.001± 0.0003 | – |
| 45* | Spain | 21 | ET-AAS | – | 1.7±0.5 | 0.07 ±0.03 | 0.1±0.05 | 0.079 ±0.07 | 0.7 ±0.22 | – | 0.006 ±0.004 | – | 0.05 ±0.051 | – |
| 46* | India | 75 | – | – | – | – | 0.1± 0.001 | 0.002± 0.001 | 4.2± 0.001 | – | 9×10⁻⁵ ± 0.002 | – | – | – |
| 47 | Islands | 18 | AAS | 0.17±0.02 | – | – | 0.1± 0.039 | – | 3.06± 0.12 | – | – | – | – | – |
| 48 | Spain | 10 | AAS | 0.3±0.03 | – | – | 0.1 ±0.012 | – | 3.6 ±0.06 | – | – | – | – | – |

Bold color is level of metals above standard limit of Internationale Standards (International Dairy Federation, FAO/WHO, and Codex Alimentarius) or standard limit of local Standards; level of As only above of China Food and Drug Administration (CFDA). AAS: Atomic Absorption Spectroscopy, MP-AES: Microwave Plasma—Atomic Emission Spectrophotometer, ICP-AES: Inductively Coupled Plasma Atomic Emission Spectroscopy, GFAAS: Graphite Furnace Atomic Absorption Spectroscopy, FAAS: Flame Atomic Absorption Spectrophotometer, ICP-MS: Inductively Coupled Plasma-Mass Spectrometry, ICP-OES: Inductively Coupled Plasma—Optical Emission Spectrometry, LIBS: Laser Induced Breakdown Spectroscopy, ICP-SFMS: Inductively Coupled Plasma Sector Field Mass Spectrometry, ASV: Anodic Stripping Voltammetry, ET-AAS: Electrothermal Atomic Absorption Spectrophotometer, PAV: polarography apparatus voltammeter; −: no data available, ND: not detected N: sample size

* the maximum values is reported, SD: standard deviation

**3.3.2. Cd.**  Cadmium (Cd) is one of the non-degradable highly toxic element with long half-life (15–30 years) [69, 73]. This metal enters the environment through industrial activities, using chemical fertilizers, pesticides, vehicle traffic, coal combustion, and using Ni-Cd rechargeable batteries [76, 77]. Cadmium first binds to blood cells and albumin and

accumulates in tissues, especially kidneys [71], liver (15%), and muscle (20%) [76]. The Itai Itai disease, hepatic, carcinogenesis, teratogenic, nephrotoxic, skeletal, and reproductive effects are detected as the multifaceted deleterious effects of this metal on the human health [70, 78]. The International Agency for Research on Cancer (IARC) has classified Cd in group 1 carcinogens for humans [73].

In this systematic review, 35(73%) studies that detected Cd in pasteurized and sterilized milk samples collected in 14 countries Pakistan [4, 11, 42, 55, 58], Iran [3, 12, 20, 23, 24, 36, 41, 43, 48, 50, 62], Iraq [39], Egypt [18, 45, 53, 56], Libya [35], China [37, 44, 49], Serbia [47], Slovakia [51], North Korea [52], Romania [25, 38], Brazil [54], Turkey [59], Spain [65, 66], and India [60] from 1999 to 2021(Table 2).

The average level of Cd in pasteurized and sterilized milk samples across globe ranged between $5\times10^{-6}$ [25] and 0.6mg/L [59] (Table 3). It should be noted that the Cd concentraten in milk samples in 23 studies (65.7%) that was conductd in different countries Iran [12, 24, 36, 41, 43, 62], Iraqi [39], Libya [35],China [37, 49], Serbia [47], Brazil [54], Egypt [18, 45, 53, 56], Slovakia [51], Pakistan [4, 42, 55, 58], Turkey [59] and Spain [66] were above standard limit of the local standrad as well as international standards such as International Dairy Federation) 0.0026 µg/g) [79], FAO/WHO and Codex Alimentarius(0.01 mg/L) [80]. This result can be due to uncontrolled and rapid industrial development in these countries cause elevated level of Cd in food stuffs [81]. It was also evidenced that Cd generally comes from phosphate fertilizers that contain up to 53.2 mg/kg of Cd [9]. Besides, equipment used in the production and packaging process is the likely other source of contamination [8, 10]. The contamination of containers and equipment used in milking and pasteurization processes and environmental contamination were accounted to be among the factors affecting the elevated Cd concentration in the collected milk samples in their study [8, 10].

From literature, highest Cd concentration(0.6±0.0001 mg/L) in milk samples was detected in Turkey [59]. The authors concluded that the high Cd content in the consumed raw milk was caused by unwanted contamination during the production process, excessive use of chemical fertilizers and pesticides, and contact with employed equipment (e.g., mechanical milkers, metal containers, etc.. Consequently, such elements as Cd present in the equipment or containers could enter raw milk during the pasteurization and sterilization process [59]. In their study, the measured Cd level exceeds the maximum Cd limit (0.01 mg/L) both in terms of the local standard, the FAO/WHO standard, and Codex Alimentarius [59].

In Slovakia, average of Cd level in sterilized milk samples was of 0.3±0.2 mg/L, indicating that Cd concentration in milk samples exceeded the local standard limit [51]. In Egypt, Salah Fathy Aal et al. (2012) claimed that a elevated level of Cd (0.3±0.01 mg/L) in sterilized milk samples, which was caused by Sn and Cd migration from milk packaging cans during the storage process [53]. Similarly, Sahar Issa et al. (2015) announced a high Cd concentration (0.29 ±0.1 mg/L) in pasteurized milk in Egypt, which exceeded the maximum standard limit and animal feed contamination with Cd was known as the main source [45].

In Iran, Cd level in pasteurized milk were ranged from 0.005±0.009 [62] mg/L to 0.02±0.04 mg/L [12], and almost 70% of the total milk samples contained highest Cd concentration higher than the permissible standard limit [12]. They reported the highest Cd level was measured in milk samples collected from caws reared close to industerial areas, mine tailings, and areas with a high traffic volume can lead to Cd accumulation in livestock tissues and, particularly, in milk [12]. In contrast, Data extracted from other studies (12 of 35 studies) were conducted in South Korea [45], Romania [19, 31], Spain [58] and India [53] (Table 2) showed minimum or no contamination of Cd above standard limit suggesting the suitable implementation of regulations in these countries.

**Table 3. Estimated Daily Intake (EDI) in of heavy metals.**

| Code | EDI (mg/kg BW/day) | | | | | | | | | | |
|---|---|---|---|---|---|---|---|---|---|---|---|
| | Fe | Al | Ni | Cu | Pb | Zn | Hg | Cd | Co | Cr | As |
| 1 | – | – | – | – | 4.29E-05 | – | – | 1.43E-05 | - | - | - |
| 2 | 7.71E-05 | - | 3.0E-06 | 4.28E-06 | 2.14E-06 | 8.61E-05 | - | - | - | 1.71E-06 | - |
| 3 | - | - | - | 1.8E-03 | 4.00E-05 | 8.6E-03 | - | 4.00E-05 | - | - | - |
| 4 | - | - | - | - | 1.60E-05 | - | - | - | - | - | - |
| 5 | - | - | - | - | 2.00E-05 | | - | - | - | - | - |
| 6 | 1.14E-03 | - | - | 5.71E-05 | - | 7.31E-03 | - | 2.28E-04 | - | 8.57E-04 | - |
| 7 | - | - | - | 9.43E-04 | 2.99E-03 | | - | - | - | - | - |
| 8 | - | - | - | - | 6.00E-06 | | - | 8.00E-07 | - | - | - |
| 9 | - | - | 1.10E-04 | 3.14E-04 | 6.29E-04 | 2.20E-03 | - | 4.71E-06 | - | 1.10E-04 | - |
| 10 | - | - | - | - | 1.03E-06 | | - | - | - | - | - |
| 11 | 1.30E-02 | 4.11E-04 | 6.17E-05 | 2.74E-05 | - | 5.48E-03 | - | - | - | 6.85E-05 | 3.36E-05 |
| 12 | 1.41E-02 | 2.39E-02 | - | 3.5E-03 | 3.57E-04 | 1.92E-02 | - | 2.85E-05 | - | 6.42E-04 | - |
| 13 | 6.2E-03 | - | - | 4.0E-04 | 2.0E-05 | | - | 1.96E-05 | - | - | 1.4E-05 |
| 14 | - | - | - | - | 4.29E-07 | | - | 3.86E-06 | - | - | - |
| 15 | - | - | - | 4.0E-04 | 8.0E-05 | 1.80E-03 | - | 9.80E-05 | - | - | - |
| 16 | - | - | - | 6.0E-04 | 1.80E-05 | 8.00E-04 | - | 2.00E-06 | - | - | - |
| 17 | 1.14E-02 | - | 7.14E-04 | 1.42E-04 | 1.35E-03 | 4.85E-02 | - | 2.85E-04 | 3.57E-04 | - | - |
| 18 | - | - | - | - | 8.00E-05 | | - | 1.20E-05 | - | - | - |
| 19 | 7.24E-03 | - | 5.57E-04 | 1.11E-03 | **1.95E-02** | 2.56E-02 | **3.78E-02** | - | 5.85E-03 | - | - |
| 20 | - | 8.57E-05 | 4.28E-05 | 3.42E-05 | 5.14E-06 | 2.4E-03 | - | 1.62E-06 | 2.57E-06 | 8.57E-06 | - |
| 21 | - | 2.2E-03 | - | 2.98E-03 | **5.5E-03** | 1.21E-02 | - | 4.55E-04 | - | - | - |
| 22 | 5.57E-03 | - | - | - | **3.9E-03** | 1.67E-02 | - | - | - | 3.34E-03 | - |
| 23 | - | - | - | 1.6E-05 | 4.00E-05 | 1.80E-04 | - | 1.00E-05 | - | - | - |
| 24 | - | - | - | 6.71E-04 | 6.04E-04 | - | - | 2.69E-05 | - | - | - |
| 25 | 1.78E-02 | - | - | 2.12E-02 | 1.6E-06 | 5.62E-02 | - | 1.38E-06 | - | - | 1.6E-06 |
| 26 | - | - | - | - | 9.29E-06 | - | - | 2.79E-06 | - | - | - |
| 27 | - | - | - | 1.20E-03 | 6.00E-06 | 1.01E-02 | - | 5.80E-07 | - | - | - |
| 28 | - | 2.0E-04 | - | - | 2.0E-05 | - | 4.0E-05 | 6.0E-06 | - | - | 4.0E-06 |
| 29 | - | - | - | 6.00E-04 | 1.80E-05 | 8.0E-04 | - | 2.0E-06 | - | - | - |
| 30 | 1.16E-02 | - | 6.2E-03 | 5.52E-03 | - | 6.99E-02 | - | **1.14E-03** | - | - | - |
| 31 | - | - | 5.71E-05 | 1.71E-04 | 5.71E-06 | 2.69E-03 | - | **1.84E-03** | 2.86E-06 | 1.71E-04 | 5.71E-07 |
| 32 | - | 2.82E-03 | - | - | 1.1E-03 | - | - | 4.71E-04 | - | - | - |
| 33 | - | - | 9.29E-06 | 1.86E-03 | 9.29E-05 | 2.97E-02 | - | 9.29E-05 | 9.29E-06 | 5.57E-04 | - |
| 34 | - | - | - | - | 1.11E-03 | 7.29E-02 | - | 3.90E-04 | - | - | - |
| 35 | 1.42E-03 | - | 4.28E-05 | 2.85E-03 | 4.28E-06 | 1.35E-02 | - | 2.14E-04 | - | 7.14E-07 | - |
| 36 | 1.09E-01 | 3.8E-02 | 4.64E-04 | 9.28E-04 | 9.28E-07 | 1.39E-02 | - | 4.64E-08 | - | 9.28E-04 | - |
| 37 | 9.42E-04 | - | 4.71E-05 | 1.57E-04 | 3.14E-05 | 4.87E-03 | - | 3.14E-05 | 4.71E-05 | 4.71E-05 | - |
| 38 | 2.81E-02 | - | - | - | - | 2.59E-02 | - | - | - | - | - |
| 39 | - | 1.28E-02 | 1.42E-03 | - | 3.57E-04 | - | - | 3.57E-04 | - | - | - |
| 40 | 2.05E-03 | - | - | 2.57E-04 | 2.57E-05 | 3.08E-03 | - | 3.42E-06 | - | 8.57E-05 | - |
| 41 | - | - | - | 1.43E-03 | - | - | - | - | - | - | - |
| 42 | 5.85E-03 | - | 3.9E-04 | 9.47E-03 | 1.11E-03 | 2.55E-02 | - | - | - | 4.40E-04 | - |
| 43 | - | - | - | 6.17E-04 | **1.99E-02** | - | - | **4.11E-03** | - | - | - |
| 44 | 7.0E-04 | 2.1E-04 | 7.0E-05 | 1.4E-04 | 1.4E-05 | 2.17E-02 | - | 3.5E-06 | - | 7.0E-06 | - |
| 45 | - | 1.19E-02 | 4.9E-04 | 7.0E-04 | 5.53E-04 | 4.9E-03 | - | 4.2E-05 | - | 3.5E-04 | - |
| 46 | - | - | - | 4.14E-04 | 8.29E-06 | 1.74E-02 | - | 3.73E-07 | - | - | - |

*(Continued)*

**Table 3.** (Continued)

| Code | EDI (mg/kg BW/day) | | | | | | | | | | |
|------|------|------|------|------|------|------|------|------|------|------|------|
| | Fe | Al | Ni | Cu | Pb | Zn | Hg | Cd | Co | Cr | As |
| 47 | 1.0E-03 | - | - | 1.0E-03 | - | 3.06E-02 | - | - | - | - | - |
| 48 | 2.1E-03 | - | - | 7.0E-04 | - | 2.52E-02 | - | - | - | - | - |

PTDI values: Fe(0.8 mg/kg BW/day); Al(0.14 mg/kg BW/day); Ni(0.3 mg/kg BW/day); Cu(0.5 mg/kg BW/day); Pb(0.0035mg/kg BW/day); Zn(1 mg/kg BW/day); Hg (0.00057 mg/kg BW/day) (for inorganic mercury); Cd(0.0008 mg/kg BW/day); Co(0.005 to 0.04); Cr(0.05 to 0.2); As(0.0021 mg/kg BW/day) [75, 124];–: no data available; The bold number shows the EDI value above the PTDI values

**3.3.3. As.** Arsenic enters the environment and water through natural and anthropogenic sources, such as tanning, ceramic, and mining industries, chemical fertilizers, and pesticides. The dissolution of As-containing mineral rocks in nature and soils naturally containing high As contents can result in its entry into the water [68, 82, 83]. Arsenic has been classified in group 1 of carcinogenic compounds by the International Agency for Research on Cancer (IARC) and EPA [84]. After entry into the body, this metal is absorbed into the bloodstream and attacks vital organs [68]. Short-term exposure to As may cause nausea and vomiting, reduced production of erythrocytes, abnormal heartbeat, irritation in hands and feet, and damage to blood vessels. Long-term exposure can produce skin lesions, various cancers, neurological problems, lung diseases, hypertension, cardiovascular diseases, infertility, miscarriage, and diabetes [85–87].

The present systematic review (Table 2) recorded As concentration measurements in pasteurized and sterilized milk samples in five (10.4%) out of 48 surveys in three countries Turkey [34], Iran [36, 41, 48], and North Korea [52] on the globe. The average level of As in pasteurized and sterilized milk samples across globe ranged between 0.0008 [48] and 0.007 mg/L [41]. The maximum acceptable As level in milk has been specified at 0.1 mg/L by EU and FAO/WHO standards [88]. In these five studies, As levels are in the safe range and were not reported beyond the local standard limits of the studied countries and EU and FAO/WHO standards. Although the number studies related to measuring As in pasteurized and sterilized milk across the world was so limited; several studies evaluated As concentration in raw cow's milk which report higher values than ours from Córdoba, Argentina with mean concentration between 0.0003 and 0.0105 mg/kg [8]; in Iran, in the range of 0.015 to 0.026 mg/kg [89], in Alabria, Italy, reporting an average As content of 0.038 mg/kg of raw milk, in Slovakia, with mean concentration of As < 0.03 mg/kg [8, 10]. All these low concentrations of As in raw cow's milk and pasteurized and sterilized milk are indicative that As use in the transformation of dairy products is safe for consumer and it does not pose health risks for human consumption.

**3.3.4. Hg.** Mercury residue enters the water through mining, different factories including papermaking plants, using fungicides, waste burning, and industrial/domestic sewage [90]. This toxic element is found in aqueous environments in the forms of metallic Hg, inorganic salt, and organic compounds with different toxicities [91]. Mercury organic compounds are more toxic than its inorganic compounds [68, 92]. The entry of the most toxic Hg form (methylmercury) into the human body causes minamata disease, which results in various neurological complications and even death [78]. The highest inorganic Hg concentration is accumulated in kidneys whereas organic Hg is more inclined to brain cells [68].

The greater Hg concentration in milk than the standard limit can be attributed to the anthropogenic contamination of the environment with Hg, Hg mining, different factories including papermaking plants, overuse of pesticides, waste burning, and industrial/urban effluents [16, 77, 90]. Consequently, this toxic element enters the bodies of lactating animals by

using contaminated fodder and water [77, 93]. Other sources of contamination with Hg in animal diets include fish meal used in animal feed [16, 19].

Mercury concentration in pasteurized and sterilized milk samples are measured in 2 studies from 2 countries (Brazil and Iran) from 2013 to 2015 [22, 36]. Data presented in Table 2 showed that limited data are available related to the prevalence of Hg in pasteurized and sterilized milk. Of 4 articles studied on Hg, in pasteurized and sterilized milk samples, Hg were detected in two studies conducted in Iran and Brazil [22, 36], with an average Hg concentration of 0.02±0.009 [36] and 6.8±3.2 [22], respectively.

A higher elevated Hg concentration (6.8±3.2 mg/L) than the FAO/WHO standard limit (0.01 mg/L) [94] was only reported in Brazil by Marcio Augusto et al. (2015) [22].

**3.3.5. Cr.** Chromium (Cr) occurs in the environment in two forms: trivalent chromium (III), it is natural and an essential nutrient occuring from litoghneic sources (e.g. plantes, animal, rocks, and volcanic gases), and hexavalent chromium (VI), which, is most commonly produced by anthropogenic activities(e.g. industrial sewage discharge, fertilizers, aircraft manufacturing, pesticides, some dyes, and the sewage of plating industries [92, 95]. The IARC reported that Cr (VI) is much more toxic than Cr (III), for both chronic and acute exposures [96, 97]. It was evidenced that the respiratory tract is the main target organ for Cr(VI) toxicity such as coughing, shortness, inhalation exposures, heezing, perforations, bronchitis, decreased pulmonary function, and pneumonia [98, 99].

Likewise, IARC have clearly noted that inhaled CR(VI) is a carcinogen element resulting in an increased lung cancer via inhalation exposure [97, 100]. Environmental Protection Agency (EPA) classified Cr (III) compounds as a non-carcinogenic element (group D), and no information are available on the reproductive or developmental effects of Cr (III) in human [101]. Based on EPA, the reference concentration for Cr (VI) is 0.0001 and 0.000008 mg/m$^3$ based on respiratory effects in rats and human, respectively [101].

In this review, the level of Cr in pasteurized and sterilized milk samples were measured in 16(33%) out of 48 studies in 10 countries Libya [21], Iraq [39], Egypt [18, 56], Turkey [34], Pakistan [4, 55], China [37, 44], Brazil [46, 64], North Korea [52], Romania [25, 38], and Spain [65, 66] across the world (Table 2).

The average level of Cr in pasteurized and sterilized milk samples across globe ranged between 0.0001 [55] and 0.6 mg/L [46]. In Brazil, a high Cr concentration(0.6±1) was reported in sterilized milk samples, which exceeded the maximum acceptable Brazilian standard(0.1 mg/L) [102].

The Cr contamination in milk samples was claimed to probably result from mines, using agricultural chemicals, diet enrichment of lactating cows with Cr-containing salts to compensate for mineral deficiencies, and Cr contamination during milk processing [46].

**3.3.6. Zn.** Zinc is essential for growth and development and plays a vital role in gene expression regulation, cell division, immune system function, and sexual maturity [103]. Zn is naturally found in soil but its concentration increases unnaturally because of anthropogenic activities mines, coal, and waste burning. Pesticides and animal manure are the other sources of contamination with this metal [78]. The main clinical symptoms of severe Zn deficiency in humans are growth retardation and delay in sexual and skeletal maturation [91, 104, 105]. This element is daily needed for the body at different concentrations, but it causes toxic effects on human health at concentrations higher than the allowable limit. The excessive use of Zn impairs Cu absorption and potentially leads to Cu deficiency [92]. There are few reports of acute Zn poisoning. Its demonstrations include nausea, vomiting, diarrhea, fever, lethargy, anemia, fatigue, and immune dysfunction [91, 105, 106].

Here, 32 (66.6%) studies measured Zn concentrations in pasteurized and sterilized milk samples in 15 countries Libya [21], Iran [12, 20, 23, 24, 48, 50, 62], Iraq [39], Egypt [18, 45, 56],

Turkey [34], Pakistan [4, 42, 55], Brazil [22, 46, 54, 64], China [37, 44], Slovakia [51], North Korea [52], Romania [25, 38], Polland [57], Spain [61, 65, 66], India [60], and Island [67] from 1993 to 2020.

The average level of Zn in milk samples across globe ranged between 0.09 [62] and 28.1 mg/L [48]. Review on Zn concentration in Milk samples showed that 20 studies out of 32 (62.5%) found exceed permission level recommended by of WHO/FAO (2.5–6.7 mg/L) [105].

A higher elevated concentration was recorded in Iran (28.1±8.32 mg/L) [48], Egypt (7.7 ±2.9 mg/L) [45], Iraq (12.8 mg/L) [39], Slovakia (11.3±1.9) [51], and Brazil (13.08±0.4 mg/L) [54].

**3.3.7. Ni.** Nickel (Ni) is an essential mineral for humans and act as a cofactor for some enzymes [77, 106]. Likewise, it is used industrial application such as steel industries and combustion of coal [68, 71]. This metal has toxic effect at concentrations higher than the maximum permissible limits and causes cell damage and changes in the activity of enzymes and hormones [68, 78, 92]. Long-term exposure damages different body organs and causes allergic skin reactions [71].

Table 2 represents the analyzed data extracted from 16(33%) out of 48 studies measured Ni in 10 countries Libya [21], Egypt [18, 56], Turkey [34], Pakistan [42, 55, 58], Brazil [22, 64], China [44], Slovakia [51], North Korea [52], Romania [25, 38] and Spain [65, 66] from 1999 to 2020.

Mean Ni concentration in pasteurized and sterilized milk samples ranges from 0.001 [38] to 1.01 mg/L [51]. According to the EU standard, the maximum acceptable limit for Ni has been determined at 0.2 mg/L [107].

A higher elevated concentration than permission limits was recorded in 4(25%) out of 16 studies from Slovakia (1.01±0.3 mg/L) [51], Pakistan (0.14±0.04–0.2±0.01 mg/L) [42, 58], and Brazil(0.15±0.3) [22], which was attributed to the use of Ni-containing vessels during transportation processes [51, 58].

**3.3.8. Co.** Cobalt (Co) is an essential element for the human, which exist in organic and inorganic forms. It acts as a an essential constituent of the B12 vitamin that plays a vital role in the formation of amino acids and some proteins in neurons, which are necessary for the proper function of the body [106]. Exposure to Co is often mixed anthropogenic activities, including the use of Co-containing sludge, phosphate fertilizers, and activities such as mining, smelting, purification, and combustion of metals [78]. This element does not accumulate in the body and is rapidly excreted through the urine as well as via feces and bile to a lesser extent [108]. Its acute effects include congestion, edema, reduced pulmonary ventilation during inhalation, and pulmonary hemorrhage during inhalation [109, 110]. The chronic inhalation of Co also causes asthma, respiratory irritation, decreased pulmonary function, pneumonia, cardiac effects, vomiting and nausea, diarrhea, liver disorders, and sensitive skin [111, 112].

In this study, six(12.5%) of 48 reviewed articles measured Co concentrations in pasteurized and sterilized milk samples in the Pakistan [42], Brazil [22], China [44], North Korea [52], Romania [38], and Egypt [56] from 2007 to 2015 (Table 2).

limited studies examined Co concentration in milk samples. The average level of Co in pasteurized and sterilized milk samples across globe ranged between 0.001±0.0001 [38] and 1.05 ±0.8 mg/L [22]. In these six studies, the Co concentration was in the immune range and was not reported beyond maximum permission level limit.

**3.3.9. Fe.** Iron is an essential element required by the human body, which is involved in important functions, including oxygen transfer in erythrocytes, helping in the correct function of enzymes, and immunity system enhancement; it also participates in DNA synthesis and electron transfer [77, 106, 113]. Iron is released by natural processes such as soil erosion, volcanic activities, and anthropogenic activities including pesticide application, mining, metal

industries., fossil fuels, industrial residues, and sewage [103]. Since Fe can produce free radicals, its excessive concentrations can cause tissue damage and increase the risk of cancer development [72, 77, 106, 113].

Here, 19 studies (39.58%) measured Fe concentrations in pasteurized and sterilized milk were measured in 13 countries Libya [21], Iraq [39], Turkey [34], Pakistan [4, 42, 55], Iran [41, 48], Brazil [22, 46, 64], Slovakia [51], Romania [25], Egypt [56], Polland [57], China [37], Spain [61, 65] and Iceland [67] from 1993 to 2020.

The average level of Fe in pasteurized and sterilized milk samples across globe ranged between 0.17±0.02 [67] and 28.1 mg/L [25]. Out of 19 studies, Fe concentrations were higher than the local standard limits only in three studies(16%) were conducted Turkey [34], Egypt [56], and China [37], and Fe concentrations were lower than the local standard limits in 16 (84%) [4, 21, 22, 25, 39, 41, 42, 46, 48, 51, 55, 57, 61, 64, 65, 67].

Fatma Esra Totan, et al (2017) and Alou Arab et al. (2007) reported that high concentration of Fe examined pasteurized and sterilized milk samples could be due to use of iron containers for milk transportation and preservation [34, 56]. Li-Qiang et al. claimed that the use of equipment with iron material for transportation and preservation of pasteurization milk could be the main pathway to increase concentration of Fe in milk samples [37].

**3.3.10. Al.** Currently, there is considerable evidence that Al may produce side effects [77]. The Fe-binding protein, called transferrin, is the main carrier of Al ions in the plasma. After absorption, Al is distributed in all body tissues, but it further accumulates in some tissues, particularly bones and lungs [114]. About 95% of Al is excreted through urine and a small amount via bile and feces [72]. The symptoms indicative of higher Al content in the human body include nausea, mouth ulcers, skin ulcers, vomiting, and diarrhea; however, these symptoms are reported to occur slightly in the short term. Al also adversely affects the nervous system, leading to memory loss, balance problems, and diseases such as Alzheimer's disease and Parkinson's dementia [72, 78].

As shown in Table 2, Al content in pasteurized and sterilized milk samples was only studied in 10 (20.83%) out of 48 studies in seven countries Turkey [34], Pakistan [4, 58], China [44], Egypt [45, 53], Iran [36], Romani [25], and Spain [65, 66] from 1999 to 2017.

The average level of Al in pasteurized and sterilized milk samples across globe ranged between 0.03 [65] and 3.35mg/L [4]. According our results, Al were higher than standards limits only in three (30%) studies were conducted in Pakistan (3.35 mg/L) [4], Egypt (1.8±0.07) [53], and Pakistan(1.8±0.13) [58].

The highest Al level (3.35 mg/L) in milk samples was observed in Pakistan, which was attributed to fresh milk contamination in aluminum containers during the transportation process [4]. It was evidenced that high concentration of Al result from the nature of using cans for pasteurized milk packaging [4, 53, 58]. Metal alloys (e.g., Al, Fe, Ni, and some other metals) used for manufacturing tools and containers may be released into milk during the pasteurization process [4]. Another reason is that Al may enter milk through milking equipment because of raw milk contact with the equipment used for the pasteurization process. The levels of some heavy metals (e.g., Al) may also increase in milk samples due to environmental contamination [53, 58].

**3.3.11. Cu.** Cu is an essential mineral for the normal growth of humans [91, 106]. Acute toxicity with Cu is associated with symptoms such as abdominal pain, nausea, vomiting, diarrhea, gastrointestinal bleeding, headache, dizziness, fever and chills, muscle pain, hemolysis of erythrocytes, anemia, liver/kidney/muscle/skeletal/ nervous problems, shock, and death [104, 115].

According to this review (Table 2), Cu concentrations in pasteurized and sterilized milk samples were measured in 34 (70.83%) out of 48 studies in 15 countries Libya [21], Iran [12, 20, 23, 24, 41, 48, 50, 62], Iraq [39], Egypt [18, 40, 45, 56], Turkey [34, 59], Pakistan [4, 42, 55,

63], Brazil [22, 64], China [37, 44], Serbia [47], Slovakia [51], North Korea [52], Romania [25, 38], Spain [61, 65, 66], India [60], Island [67] from 1993 to 2020.

The average level of Cu in pasteurized and sterilized milk samples across globe ranged between 0.004 [34] and 10.6 mg/L [48]. It should be noted that 32 studies out of 34(94%) found to exceed the maximum limit of Cu in milk samples recommended by Standardization Administration of the People's Republic of China (0.01 mg/L) [116] and WHO/FAO (0.05 mg/L) [117]. A higher elevated concentration was recorded in Iran(10.6±5.32 mg/L) [48], Egypt (1.9±0.7 mg/L) [45], and Brazil (1.7±0.8 mg/L) [64].

The high Cu concentration was attributed to environmental contamination [23, 37, 45]. Conversely, much lower level of Cu (Cu< 0.01 mg/l) were detected in only 2 studies were conducted in Iran and Turkey in 2014 [34, 62].

Based on result of this review, about the prevalence of heavy metals in the pasteurized and sterilized milk samples, it is clear that Cu, Pb, Zn, and Cd were the predominant heavy metal among all the milk samples. This may be due to the way of these heavy metals excretion in milk samples depending quality of raw milk, manufacturing procedures, as well as heavy metal's non-biodegradable properties and persistent nature after being accumulated and absorbed [13, 17, 118].

High concentration of heavy metals in raw milk may be attributed to contaminated animal feed and drinking water. Likewise, the mal-usage of fertilizers can affect the heavy metals uptake by the plants, which are consumed by the animals [118–121]. It is evidenced that the evaporation of milk during pasteurization process generally increase contaminant concentration in milk and its products. Furthermore, heavy metals including, Cd, Cu Pb, and Cr, preferentially bound to milk caseins, and improper production methods in unhygienic atmospheric contamiantedconditions and poor packaging can transfer heavy metals to milk and its products [104, 122, 123].

## 3.4. Risk assessment of pasteurized and sterilized milk consumption

**3.4.1. Estimated Daily Intake (EDI).** Dietary exposure to Fe, Al, Ni, Cu, Pb, Zn, Hg, Cd, Co, Cr and As through pasteurized and sterilized milk consuming was evaluated by calculating EDI of these metals based on average of heavy metals in this review (Table 3). The value of EDI was compared with the provisional tolerable daily intake (PTDI) to appraise the risk associated with ingesting heavy metals in milk.

According our finding, some studies reported that the EDI values of Pb [22, 45, 46, 59], Cd [51, 52, 59] and Hg [22], higher than PTDI limits, while others trace elements such as Fe, Al, Ni, Cu, Zn, Co, Cr, and As for all examined samples did not exceed the PTDI limits.

The Joint FAO/WHO Expert Committee on Food Additives recommended the provisional tolerable intakes of Pb as(0.0036 mg/kg BW/day) [75]. Daily intake for the exposure of Pb through Pasteurized and sterilized milk consumption around the world ranged from 4.29E-07 [35] to 1.99E-02 [59] (mg/kg BW/day) which represent 0.012%–568.16% of PTDI [35, 59].

It should be noted that mean Pb uptake by milk consumption in 4(8.33%) out of 48 studies across the globe were extremely high and it was exceeded 100% of PTDI [22, 45, 46, 59]. The average consumption of Pb through milk reaches its maximum in milk collected in Turkey [59], Brazil [22, 46] and Egypt [45], the values were 568.16%, 557.14%, and 157.14% of PTDI respectively (Table 4), which indicated the high potential health risk from Pb tested upon consumption in these milk samples. In 16 (39.02%) studies the values were between 1% to 93% of PTDI. In 21 studies (51.21%) were representing values <1% of PTDI.

For Cd, the PTMI is 0.00083 mg/kg BW/day [75]. Cadmium uptake by milk consumption observed in 36 (75%) out of 48 studies. The exposure of Cd through Pasteurized and sterilized

milk consumption across the globe ranged between 4.64E-08 [25] to 4.11E-03 [59] (mg/kg BW/day) which represented the values ranged from 0.0055% [25] to 495.69% [59] of PTDI. The average consumption of Cd through milk collected in Turkey (495.69%) [59] and Slovakia (222.03%) [51] covers maximum values and of PTDI. The average Cd consumption exceed 100% of PTDI in 2 Studies across the globe [51, 59], while the values ranged between 1% to 90% of PTDI in 18 Studies (51.42%), and 15 studies (42.85%) were representing values <1% of PTDI (Table 3).

Regarding Hg, a recommended PTDI of 0.00057 mg/kg BW/day have been set by the 72th JECFA(the Joint FAO/WHO Expert Committee on Food Additives) in 2010 form inorganic mercury [75]. It should be noted that Hg uptake by milk consumption only observed in 2 (4.1%) out of 48 studies [22, 36]. The average of Hg consumption through milk ranged between 4.0E-05 to 3.78E-02 (mg/kg BW/day) which represent 7.01% to 6646.61% of PTDI. The average consumption of Hg in sterilized milk collected from Brazil (6646.61%) covers maximum value of PTDI [22] (Table 3). Therefore, it is seem that exposure to Pb [22, 45, 46, 59], Cd [51, 52, 59], and Hg [22], through milk consumption contributed the highest portion of EDI.

**3.4.2. Non-carcinogenic risk assessment.** The non-carcinogenic risk of heavy metals for the milk consumers was determined by calculating the target hazard quotient (THQ) and hazard index (HI) values (Table 4 and Fig 2). If THQ or HI of heavy metals is lower than the threshold of the US EPA criteria (HQ< 1), risk is improbable to happen. Non-carcinogenic effect can occur to exposed population if THQ is higher than the threshold of the US EPA criteria (HQ> 1) [105].

In milk samples, the average value of THQ for most heavy metals were found to be lower than the threshold of the US EPA criteria, except for Pb [22, 45, 46, 59], Hg [22], Cd [51, 59]) and Co [22].

Therefore, non-carcinogenic risk is expected for Pb, Hg, Co, and Cd. Review on THQ values showed that milk consumers in 4 studies out of 48 in the Brazil [22, 46], Egypt [45], Turkey [59] were exposed to high health risk through the intake of Pb. The highest value of THQ for Pb was estimated to be 5.68E+00 in pasteurized and sterilized milk collected in Turkey [59]. Also, THQ of Pb was more than one in milk samples collected from Brazil(from 1.11E+00 to 5.57E+00) [22, 46] and Egypt (1.57E+00) [45].

Review on the THQ of Cd showed that milk consumers in 2 studies was conducted in the Slovakia [51] and Turkey [59] were exposed to high health risk through the intake of Cd. The highest THQ value of Cd was 4.11E+00 recorded in pasteurized and sterilized milk collected in Turkey [59] in 2002. Also, HQ values were higher than 1 in sterilized milk collected in Slovakia (1.84E+00) [51] in 2012.

For Hg and Co, THQ values of milk samples exceeded 1 in only one study were conducted in Brazil [22]. The results of non-carcinogenic risks from exposure to metals through milk consumption indicates that Pasteurized and sterilized milk collected in all sites during the last decade was safe for human consumption in terms of the amounts of Fe, Al, Cu, Ni, Zn, Cr and As (HQ values <1; Table 4).

Based on risk results, higher THQ value of >1 was detected for Pb, Cd, Co and Hg in pasteurized and sterilized milk samples examined in Slovakia [51], Brazil [22, 46], Turkey [59], and Egypt [45]. It seems that milk consumers in these regions can be exposed to potential health risks upon milk consumption. Further attention must be paid to control contamination sources of these metals.

**3.4.3. Carcinogenic risk assessment of heavy metals.** Table 5 and Fig 3 represents the carcinogenic risk (CR) of heavy metals in pasteurized and sterilized milk samples in different countries around the world. According to EPA, if CR of metals is Less than $10^{-6}$, carcinogenic

**Table 4. Target hazard quotient (THQ) and hazard index (HI) values for heavy metals in pasteurized and sterilized milk.**

| Code | THQ | | | | | | | | | | | TTHQ (HI) |
|---|---|---|---|---|---|---|---|---|---|---|---|---|
| | Fe | Al | Ni | Cu | Pb | Zn | Hg | Cd | Co | Cr | As | |
| 1 | - | - | - | - | 1.22E-02 | - | - | 1.43E-02 | - | - | - | 2.65E-02 |
| 2 | 1.10E-04 | - | 1.50E-05 | 1.07E-04 | 6.12E-04 | 2.87E-04 | - | - | - | 1.14E-06 | - | 1.13E-03 |
| 3 | - | - | - | 4.50E-02 | 1.14E-02 | 2.87E-02 | - | 4.00E-02 | - | - | - | 1.25E-01 |
| 4 | - | - | - | - | 4.57E-03 | - | - | - | - | - | - | 4.57E-03 |
| 5 | - | - | - | - | 5.71E-03 | - | - | - | - | - | - | 5.71E-03 |
| 6 | 1.64E-03 | - | - | 1.43E-03 | - | 2.44E-02 | - | 2.29E-01 | - | 5.71E-04 | - | 2.57E-01 |
| 7 | - | - | - | 2.36E-02 | 8.53E-01 | - | - | - | - | - | - | 8.77E-01 |
| 8 | - | - | - | - | 1.71E-03 | - | - | 8.00E-04 | - | - | - | 2.51E-03 |
| 9 | - | - | 5.50E-04 | 7.86E-03 | 1.80E-01 | 7.33E-03 | - | 4.71E-03 | - | 7.33E-05 | - | 2.00E-01 |
| 10 | - | - | - | - | 2.94E-04 | - | - | - | - | - | - | 2.94E-04 |
| 11 | 1.86E-02 | 5.88E-04 | 3.09E-04 | 6.86E-04 | - | 1.83E-02 | - | - | - | 4.57E-05 | 1.12E-01 | 1.51E-01 |
| 12 | 2.02E-02 | 3.42E-02 | - | 8.75E-02 | 1.02E-01 | 6.40E-02 | - | 2.86E-02 | - | 4.29E-04 | - | 3.37E-01 |
| 13 | 8.86E-03 | - | - | 1.00E-02 | 5.71E-03 | - | - | 1.96E-02 | - | - | 4.67E-02 | 9.08E-02 |
| 14 | - | - | - | - | 1.22E-04 | - | - | 3.86E-03 | - | - | - | 3.98E-03 |
| 15 | - | - | - | 1.00E-02 | 2.29E-02 | 6.00E-03 | - | 9.80E-02 | - | - | - | 1.37E-01 |
| 16 | - | - | - | 1.50E-02 | 5.14E-03 | 2.67E-03 | - | 2.00E-03 | - | - | - | 2.48E-02 |
| 17 | 1.63E-02 | - | 3.57E-03 | 3.57E-03 | 3.88E-01 | 1.62E-01 | - | 2.86E-01 | 1.00E-01 | - | - | **1.85E+00** |
| 18 | - | - | - | - | 2.29E-02 | - | - | 1.20E-02 | - | - | - | 3.49E-02 |
| 19 | 1.03E-02 | - | 2.79E-03 | 2.79E-02 | **5.57E+00** | 8.54E-02 | **9.47E+01** | - | **1.46E+01** | - | - | **1.15E+02** |
| 20 | - | 1.22E-04 | 2.14E-04 | 8.57E-04 | 1.47E-03 | 8.00E-03 | - | 1.63E-03 | 6.43E-03 | 5.71E-06 | - | 1.87E-02 |
| 21 | - | 3.14E-03 | - | 7.46E-02 | **1.57E+00** | 4.03E-02 | - | 4.56E-01 | - | - | - | **2.15E+00** |
| 22 | 7.96E-03 | - | - | - | **1.11E+00** | 5.57E-02 | - | - | - | 2.23E-03 | - | **1.18E+00** |
| 23 | - | - | - | 4.00E-04 | 1.14E-02 | 6.00E-04 | - | 1.00E-02 | - | - | - | 2.24E-02 |
| 24 | - | - | - | 1.68E-02 | 1.73E-01 | - | - | 2.69E-02 | - | - | - | 2.16E-01 |
| 25 | 2.54E-02 | - | - | 5.30E-01 | 4.57E-04 | 1.87E-01 | - | 1.38E-03 | - | - | 5.33E-03 | 7.50E-01 |
| 26 | - | - | - | - | 2.65E-03 | - | - | 2.79E-03 | - | - | - | 5.44E-03 |
| 27 | - | - | - | 3.00E-02 | 1.71E-03 | 3.37E-02 | - | 5.80E-04 | - | - | - | 6.60E-02 |
| 28 | - | 2.86E-04 | - | - | 5.71E-03 | - | 1.00E-01 | 6.00E-03 | - | - | 1.33E-02 | 1.25E-01 |
| 29 | - | - | - | 1.50E-02 | 5.14E-03 | 2.67E-03 | - | 2.00E-03 | - | - | - | 2.48E-02 |
| 30 | 1.67E-02 | - | 3.10E-02 | 1.38E-01 | - | 2.33E-01 | - | **1.84E+00** | - | - | - | **2.26E+00** |
| 31 | - | - | 2.86E-04 | 4.29E-03 | 1.63E-03 | 8.95E-03 | - | 1.14E-03 | 7.14E-03 | 1.14E-04 | 1.90E-03 | 2.55E+-2 |
| 32 | - | 4.04E-03 | - | - | 3.14E-01 | - | - | 4.71E-01 | - | - | - | 7.90E-01 |
| 33 | - | - | 4.64E-05 | 4.64E-02 | 2.65E-02 | 9.90E-02 | - | 9.29E-02 | 2.32E-02 | 3.71E-04 | - | 2.88E-01 |
| 34 | - | - | - | - | 3.18E-01 | 2.43E-01 | - | 3.90E-01 | - | - | - | 9.51E-01 |
| 35 | 2.04E-03 | - | 2.14E-04 | 7.14E-02 | 1.22E-03 | 4.52E-02 | - | 2.14E-01 | - | 4.76E-07 | - | 3.34E-01 |
| 36 | 1.57E-01 | 5.44E-02 | 2.32E-03 | 2.32E-02 | 2.65E-04 | 4.64E-02 | - | 4.64E-05 | - | 6.19E-04 | - | 2.84E-01 |
| 37 | 1.35E-03 | - | 2.36E-04 | 3.93E-03 | 8.98E-03 | 1.62E-02 | - | 3.14E-02 | 1.18E-01 | 3.14E-05 | - | 1.8E-01 |
| 38 | 4.02E-02 | - | - | - | - | 8.66E-02 | - | - | - | - | - | 1.27E-01 |
| 39 | - | 1.84E-02 | 7.14E-03 | - | 1.02E-01 | - | - | 3.57E-01 | - | - | - | 4.85E-01 |
| 40 | 2.94E-03 | - | - | 6.43E-03 | 7.35E-03 | 1.03E-02 | - | 3.43E-03 | - | 5.71E-05 | - | 3.05E-02 |
| 41 | - | - | - | 3.57E-02 | - | - | - | - | - | - | - | 3.57E-02 |
| 42 | 8.36E-03 | - | 1.95E-03 | 2.37E-01 | 3.18E-01 | 8.52E-02 | - | - | - | 2.93E-04 | - | 6.51E-01 |
| 43 | - | - | - | 1.54E-02 | **5.68E+00** | - | - | **4.11E+00** | - | - | - | **9.81E+00** |
| 44 | 1.00E-03 | 3.00E-04 | 3.50E-04 | 3.50E-03 | 4.00E-03 | 7.23E-02 | - | 3.50E-03 | - | 4.67E-06 | - | 8.50E-02 |
| 45 | - | 1.70E-02 | 2.45E-03 | 1.75E-02 | 1.58E-01 | 1.63E-02 | - | 4.20E-02 | - | 2.33E-04 | - | 2.12E-01 |
| 46 | - | - | - | 1.04E-02 | 2.37E-03 | 5.80E-02 | - | 3.73E-04 | - | - | - | 7.11E-02 |

*(Continued)*

**Table 4.** (Continued)

| Code | THQ | | | | | | | | | | | TTHQ (HI) |
|------|-----|-----|-----|-----|-----|-----|-----|-----|-----|-----|-----|-----------|
|      | Fe | Al | Ni | Cu | Pb | Zn | Hg | Cd | Co | Cr | As |  |
| 47 | 1.43E-03 | - | - | 2.50E-02 | - | 1.02E-01 | - | - | - | - | - | 1.28E-01 |
| 48 | 3.00E-03 | - | - | 1.75E-02 | - | 8.40E-02 | - | - | - | - | - | 1.05E-01 |

–: no data available; the red color shows the THQ value above the recommended value of EPA.

adverse effect are improbable to the exposed population. Carcinogenic adverse effect may happen to exposed population if CR higher than $10^{-6}$. Cancer risk was measured for 44 studies included in a systematic review (Table 5).

Based on risk results, 11(25%)studies in Libya [21], Iran [3, 5, 20, 23, 32, 50], Indonesia [33], China [44], Romania [25] and India [60] indicated acceptable or negligible levels of heavy metals (CR <10−6) via consumption of milk samples, and 25(56.8%) studies reported moderate level of risk ($10^{-4}<CR \leq 10^{-6}$) in Pakistan [4, 11, 55], Iran [12, 24, 36, 41, 43, 48, 62], Iraqi [39], Egypt [18, 40, 56], Turkey [34], Libya [35], Brazil [46, 64], Serbia [47], China [37, 49], South Korea [52], Romania [38], and Spain [65, 66].

Likewise, 8 included studies indicated high level of risk (CR≥$10^{-4}$) via consumption of milk samples contaminated with heavy metals compound in Pakistan [42], Brazil [22], Egypt [45, 53], Slovakia [51], Brazil [54], Pakistan [58], and Turkey [59].

Review on CR values showed that milk consumers in 14 studies (31.8%) out of 44 studies in the Egypt [18, 40, 45, 53]), Pakistan [4, 42, 58], Brazil [22, 46, 54, 64], Serbia [47], and Spain [66] were exposed to moderate or high carcinogenic risk through the intake of Pb. The highest

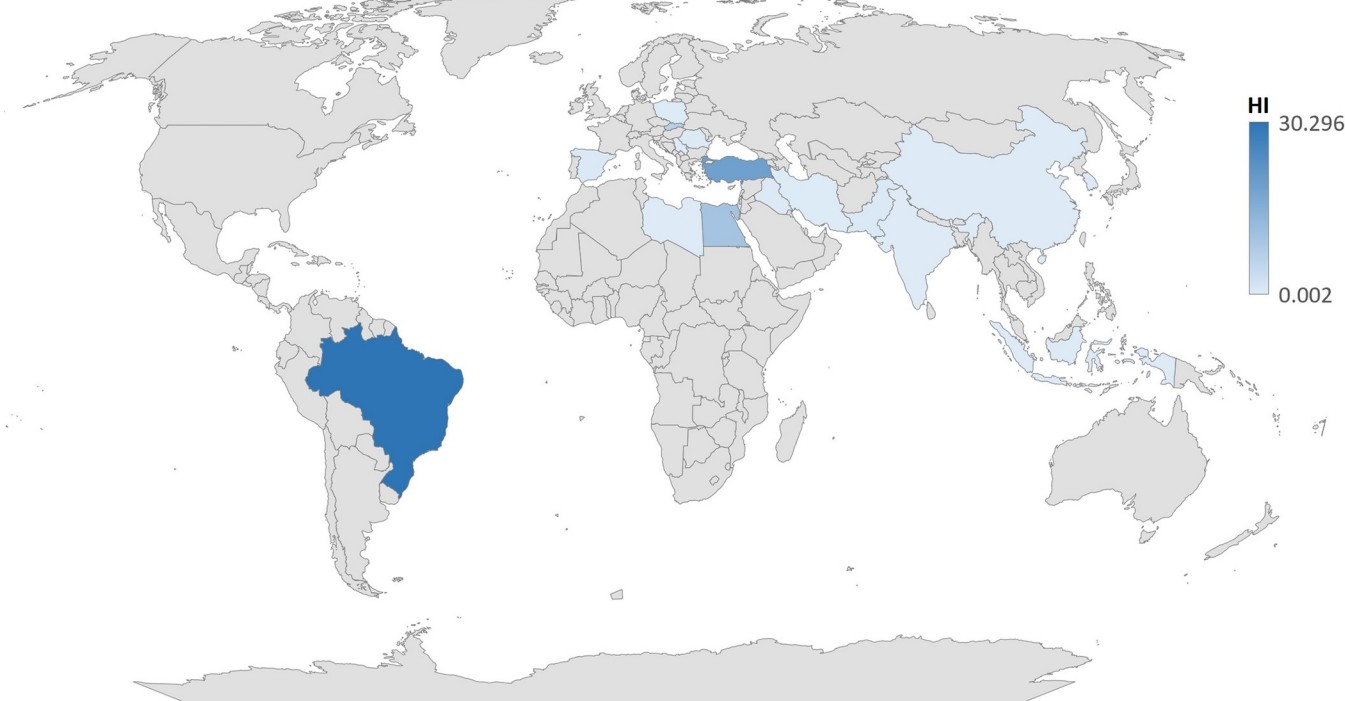

**Fig 2. Global map for hazard index values for heavy metals in pasteurized and sterilized milk (countries with no data are shown in grey).**

**Table 5. Carcinogenic risk assessment values for heavy metals in pasteurized and sterilized milk reported in research articles published since 1993–2021.**

| Code | CR | | | | TCR |
|------|------|------|------|------|------|
| | As | Hg | Pb | cd | |
| 1 | - | - | 3.64E-07 | 5.43E-06 | 5.79E-06 |
| 2 | - | - | 1.82E-08 | - | 1.82E-08 |
| 3 | - | - | 3.40E-07 | 1.52E-05 | 1.55E-05 |
| 4 | - | - | 1.36E-07 | - | 1.36E-07 |
| 5 | - | - | 1.70E-07 | - | 1.70E-07 |
| 6 | - | - | - | 8.69E-05 | 8.69E-05 |
| 7 | - | - | 2.54E-05 | - | 2.54E-05 |
| 8 | - | - | 5.10E-08 | 3.04E-07 | 3.55E-07 |
| 9 | - | - | 5.34E-06 | 1.79E-06 | 7.13E-06 |
| 10 | - | - | 8.74E-09 | - | 8.74E-09 |
| 11 | 5.04E-05 | - | - | - | 5.04E-05 |
| 12 | - | - | 3.04E-06 | 1.09E-05 | 1.39E-05 |
| 13 | 2.10E-05 | - | 1.70E-07 | 7.45E-06 | 2.86E-05 |
| 14 | - | - | 3.64E-09 | 1.47E-06 | 1.47E-06 |
| 15 | - | - | 6.80E-07 | 3.72E-05 | 3.79E-05 |
| 16 | - | - | 1.53E-07 | 7.60E-07 | 9.13E-07 |
| 17 | - | - | 1.15E-05 | 1.09E-04 | 1.21E-04 |
| 18 | - | - | 6.80E-07 | 4.56E-06 | 5.24E-06 |
| 19 | - | 1.14E-05 | 1.66E-04 | - | 1.77E-04 |
| 20 | - | - | 4.37E-08 | 6.19E-07 | 6.63E-07 |
| 21 | - | - | 4.68E-05 | 1.73E-04 | 2.20E-04 |
| 22 | - | - | 3.32E-05 | - | 3.32E-05 |
| 23 | - | - | 3.40E-07 | 3.80E-06 | 4.14E-06 |
| 24 | - | - | 5.14E-06 | 1.02E-05 | 1.53E-05 |
| 25 | 2.40E-06 | - | 1.36E-08 | 5.24E-07 | 2.94E-06 |
| 26 | - | - | 7.89E-08 | 1.06E-06 | 1.14E-06 |
| 27 | - | - | 5.10E-08 | 2.20E-07 | 2.71E-07 |
| 28 | 6.00E-06 | 1.20E-08 | 1.70E-07 | 2.28E-06 | 8.46E-06 |
| 29 | - | - | 1.53E-07 | 7.60E-07 | 9.13E-07 |
| 30 | - | - | - | 7.00E-04 | 7.00E-04 |
| 31 | 8.57E-07 | - | 4.86E-08 | 4.34E-07 | 1.34E-06 |
| 32 | - | - | 9.35E-06 | 1.79E-04 | 1.88E-04 |
| 33 | - | - | 7.89E-07 | 3.53E-05 | 3.61E-05 |
| 34 | - | - | 9.47E-06 | 1.48E-04 | 1.57E-04 |
| 35 | - | - | 3.64E-08 | 8.14E-05 | 8.14E-05 |
| 36 | - | - | 7.89E-09 | 1.76E-08 | 2.55E-08 |
| 37 | - | - | 2.67E-07 | 1.19E-05 | 1.22E-05 |
| 39 | - | - | 3.04E-06 | 1.36E-04 | 1.39E-04 |
| 40 | - | - | 2.19E-07 | 1.30E-06 | 1.52E-06 |
| 42 | - | - | 9.47E-06 | - | 9.47E-06 |
| 43 | - | - | 1.69E-04 | 1.56E-03 | 1.73E-03 |
| 44 | - | - | 1.19E-07 | 1.33E-06 | 1.45E-06 |
| 45 | - | - | 4.70E-06 | 1.60E-05 | 2.07E-05 |
| 46 | - | - | 7.04E-08 | 1.42E-07 | 2.12E-07 |

−: no data available.

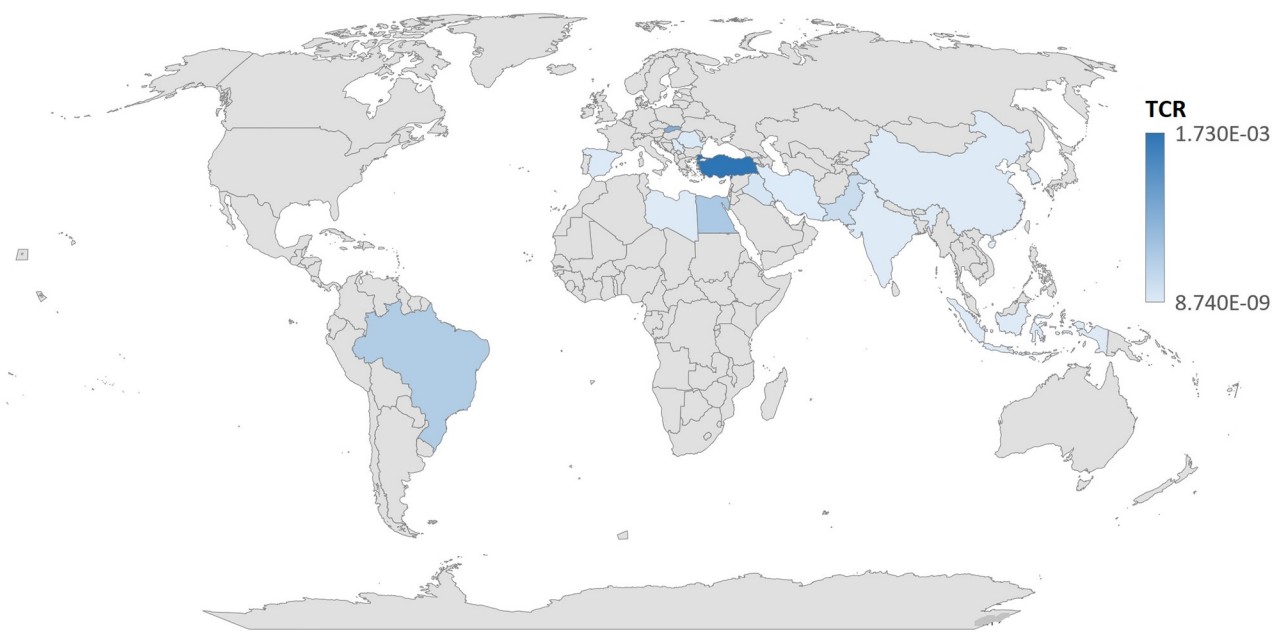

**Fig 3. Global map for the total carcinogenic risk (CR) values for heavy metals in pasteurized and sterilized milk (countries with no data are shown in grey).**

value of cancer risk for Pb was measured to be 1.66E-04 and 1.69E-04 in pasteurized and sterilized milk collected from Brazil [22] and Turkey [59], respectively.

About Cd, 35 (73%) of 48 studies included in this review measured Cd concentration in milk samples. They reported milk consumer in 26 studies (74%) out of 35 studies in Pakistan [4, 11, 42, 55, 58], Iran [12, 24, 36, 41, 43, 62], Iraqi [39], Egypt [18, 45, 53, 56], Libya [35], Serbia [47], China [37, 49], Slovakia [51], Romania [38], Brazil [54], Turkey [59], and Spain [65, 66] were exposed to moderate to high carcinogenic risk through the intake of Cd. The highest value of cancer risk for Cd was 7.00E-04 in pasteurized and sterilized milk samples collected from Slovakia [51].

Arsenic and Hg were reported only by 5 and 2 studies, respectively. They reported milk consumer in these studies were exposed to moderate carcinogenic risk through the intake of Hg and As. Although, limited studies measured As and Hg contamination in milk, these element could be a growing public concern in world.

The high exposure to heavy metals contamination especially Pb and Cd is certain public health concern, therefore, maximum control measures and stricter regulation must be adopted to chemical contaminants in the dairy industry, becuase that milk is the main products by humans worldwide.

## 4. Limitation

Although attempts were made to conduct comprehensive search strategies to include all published articles on heavy metal contamination in milk, some surveys may be lost unintentionally. Some findings like dissertations, books, and conference proceedings were not included in this study because of the low quality of results adopted in this systematic review. Given the high heterogeneity in the data, results were not pooled in the meta-analysis and only were interpreted based on the systematic review. The meta-analytical results could be obscured and meaningless, when data are too heterogeneous, or bias is present in studies. Further, the results

of this study could not lead to actual conclusions about sources of heavy metal concentration in the pasteurized and sterilized milk because 77.1% of studies did not report possible sources of heavy metal.

## 5. Conclusion

This review is the first study that was conducted on the concentration and potential health risks of heavy metals in pasteurized and sterilized milk recorded across the world. Based on the result of this review, Cu, Cd, Zn, and Pb were the most common heavy metals, which exceeded the maximum permissible criteria in 94%, 67%, 62%, and 46% of the milk samples analyzed, respectively followed by Al, Fe, and Ni, which exceeded the local maximum permissible criteria in 30%, 16%, and 12.5% of the milk tested, respectively. Arsenic, Co, and Hg showed the lowest contamination levels and were lower than the maximum permissible limits recommended by local and international standards. Since THQ results for Pb, Hg, Co, and Cd were higher than one in milk samples examined in Slovakia, Brazil, Turkey, and Egypt, milk consumers in these countries would be exposed to non-carcinogenic risk through the continuous use of pasteurized and sterilized milk. Relying on our CR results, milk consumers in 33 (68.75%) studies were exposed to moderate to high levels of carcinogenic risk through the intake of Cd and Pb, and the highest level of carcinogenic risk was reported in Pakistan, Egypt, Slovakia, Brazil, and Turkey. The high exposure to heavy metals' contamination especially Pb and Cd is a certain public health concern, therefore, maximum control measures and stricter regulation must be adopted for chemical contaminants in the dairy industry.

The presence of heavy metals contaminants in the pasteurized and sterilized milk show that they are currently used illegally in both animal husbandry and agriculture. Likewise, pasteurization and sterilization processes are not efficient for the elimination or degradation of the heavy metals addressed. In this sense, strict food safety laws should be made mandatory at every stage of milk processing units and handling to regulate and eliminate the prevalence of heavy metals under permissible limits. Further studies are required to develop safe milk processing and handling methods for the sustainable reduction of these toxic metals in milk and its products, particularly in Pakistan, Egypt, Slovakia, Brazil, and Turkey, are crucial.

## Supporting information

**S1 Checklist. PRISMA checklist.**
(DOCX)

**S1 Table. Quality assessments of the included studies based on Newcastle–Ottawa scale.**
(DOC)

## Acknowledgments

The author wishes to express her gratitude towards the vice president of research in Mashhad University of Medical Sciences.

## Author Contributions

**Data curation:** Zahra Alinezhad, Mohammad Hashemi, Seyedeh Belin Tavakoly Sany.

**Investigation:** Zahra Alinezhad, Mohammad Hashemi, Seyedeh Belin Tavakoly Sany.

**Methodology:** Zahra Alinezhad, Mohammad Hashemi, Seyedeh Belin Tavakoly Sany.

**Supervision:** Seyedeh Belin Tavakoly Sany.

**Validation:** Zahra Alinezhad, Mohammad Hashemi, Seyedeh Belin Tavakoly Sany.

**Visualization:** Mohammad Hashemi.

**Writing – original draft:** Zahra Alinezhad.

**Writing – review & editing:** Seyedeh Belin Tavakoly Sany.

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
