## [Decision Letter · Decision Letter 0]

10 Oct 2023

PONE-D-23-30074Concentration of Heavy Metals in Pasteurized and sterilized Milk and Health Risk Assessment across the Globe: A Systematic ReviewPLOS ONE

Dear Dr. Tavakoly Sany,

Thank you for submitting your manuscript to PLOS ONE. After careful consideration, we feel that it has merit but does not fully meet PLOS ONE’s publication criteria as it currently stands. Therefore, we invite you to submit a revised version of the manuscript that addresses the points raised during the review process.

ACADEMIC EDITOR:In addition to the reviewer comments, I have attached the main manuscript file with some highlights of mistakes identified in the submission. Because this is a systematic review, was a protocol for it registered with any body e.g. PROSPERO? If so, provide the link to the registered protocol.The provided PRISMA 2020 checklist appears largely edited (some items have been omitted). Please use the original checklist,and whatever is not appliocable to your study could be indicated as ''N/A'' for Not Applicable.You should synthesize your data and ensure that you indicate the implications of your study. Please submit your revised manuscript by Nov 24 2023 11:59PM. If you will need more time than this to complete your revisions, please reply to this message or contact the journal office at plosone@plos.org. Please include the following items when submitting your revised manuscript:A rebuttal letter that responds to each point raised by the academic editor and reviewer(s). You should upload this letter as a separate file labeled 'Response to Reviewers'.A marked-up copy of your manuscript that highlights changes made to the original version. You should upload this as a separate file labeled 'Revised Manuscript with Track Changes'.An unmarked version of your revised paper without tracked changes. You should upload this as a separate file labeled 'Manuscript'.

We look forward to receiving your revised manuscript.

Kind regards,

Timothy Omara, PhD

Academic Editor

PLOS ONE

Journal Requirements:

Reviewers' comments:

Reviewer's Responses to Questions

**Comments to the Author**

1. Is the manuscript technically sound, and do the data support the conclusions?

Reviewer #1: Partly

Reviewer #2: Yes

2. Has the statistical analysis been performed appropriately and rigorously? 

Reviewer #1: N/A

Reviewer #2: Yes

3. Have the authors made all data underlying the findings in their manuscript fully available?

Reviewer #1: Yes

Reviewer #2: Yes

4. Is the manuscript presented in an intelligible fashion and written in standard English?

Reviewer #1: Yes

Reviewer #2: Yes

5. Review Comments to the Author

Reviewer #1: The article is very neat and orderly, and sufficient and accurate sources have been selected.

Only a few points can be corrected.

Do not use we Example: Line 153, “we used,” Line 156,” we included”, Line 320, “we retrieved,” Line 154“In this review we” should be omitted.

Tables must be modified. It should have three main lines and numbers.

Symbols should be used instead of color numbers and explained at the end of the table.

Why didn't you do a meta-analysis? It can help the decision-making power to solve this problem and provide better results. The following article is one of the examples of Iran that is recommended to be used, which was not seen in your sources.

A systematic review and meta-analysis of lead and cadmium concentrations in cow milk in Iran and human health risk assessment. Environmental Science and Pollution Research. 2020 Apr;27:10147-59.

Reviewer #2: General comment: General comment: An important systematic review is presented on the presence of the main heavy metals in pasteurized and sterilized milk in the world, indicating that the estimated risk values range from moderate to high depending on the regions and conditions of the process and the system. of milk production and its productive conditions. It would be good if you included a text about the limitations of the study.

Minor Reviion:

In Abstract: in objective, indicate the years selected in the review, from that year until September 2023.

In Abstract: In the conclusion it would be important to list the first three countries or places in the world with the greatest health risk due to the consumption of pasteurized and sterilized milk.

Introduction: First paragraph, it is better to consider the term food safety instead of food security, because food security in terms of public nutrition is linked to the availability, access and use of food, in this way we avoid confusion among readers.

Introduction: The nutritional importance of milk is indicated (essential fatty acids, amino acids and vitamins, etc.). I also suggest summarizing its importance from the point of view of health actions. There is important information on its protective action against coronary heart disease, stroke, diabetes and hypertension, bone health, obesity, osteoporosis, decreased risk of colorectal and bladder cancer (https://doi.org/10.3390/toxics11100809; https:/ /doi.org/10.3168/jds.2020-18296).

The last line of item 2.1 is incomplete, please complete the idea.

In 3.3.2: There are works that indicate high levels of Cd due to the use of phosphorus fertilizers in pastures and passing into milk and exceeding permissible limits: (https://doi.org/10.3390/toxics11100809;
https://doi.org/10.3390/toxics11100809;
https://doi.org/10.3390/toxics11100809; .org/10.1007/s12011-023-03838-2).

In 3.3.3: If you see fit, you could report the As levels reported in the articles indicated for 3.3.2

In 3.4.2 standardize acronyms, either use TQT or just HQ.

6. PLOS authors have the option to publish the peer review history of their article (what does this mean?). If published, this will include your full peer review and any attached files.

Reviewer #1: No

Reviewer #2: **Yes: **Jorge Castro Beriñana

---

## [Author Response · Author response to Decision Letter 0]

16 Nov 2023

Academic editor’s comments

 We have improved the manuscript based on the final comments of the reviewers, to whom we sincerely thank them, as their comments have improved the quality of the article.

Because this is a systematic review, was a protocol for it registered with any body e.g. PROSPERO? If so, provide the link to the registered protocol. No, this is not registered in PROSPERO because the review process in Prospero take an average 6 months.

The provided PRISMA 2020 checklist appears largely edited (some items have been omitted). Please use the original checklist,and whatever is not appliocable to your study could be indicated as ''N/A'' for Not Applicable. It was corrected based on this comment.

You should synthesize your data and ensure that you indicate the implications of your study. We analysied our data based on measearing health risk (carcinogenic and non-carcionogenic risk) via EPA model (Table1). Given the high heterogeneity in the data, results were not pooled in the meta-analysis and only were interpreted based on the systematic review. Meta- analysis is not approprite when data are too heterogeneous, or bias is present in studies because the meta-analytical results coud be obscured and meaningless. 

The implication of this review is to provide a comprehensive report on the occurrence, concentration, and potential health risk of selected heavy metals in pasteurized and sterilized milk recorded across the world in order to compare metal concentration and risk with the permissible limits. So, the conclusion was rewritten as follows:

This review is the first study that was conducted on the concentration and potential health risks of heavy metals in pasteurized and sterilized milk recorded across the world. Based on the result of this review, Cu, Cd, Zn, and Pb were the most common heavy metals, which exceeded the maximum permissible criteria in 94%, 67%, 62%, and 46% of the milk samples analyzed, respectively followed by Al, Fe, and Ni, which exceeded the local maximum permissible criteria in 30%, 16%, and 12.5% of the milk tested, respectively. Arsenic, Co, and Hg showed the lowest contamination levels and were lower than the maximum permissible limits recommended by local and international standards. 

Since THQ results for Pb, Hg, Co, and Cd were higher than one in milk samples examined in Slovakia, Brazil, Turkey, and Egypt, milk consumers in these countries would be exposed to non-carcinogenic risk through the continuous use of pasteurized and sterilized milk. Relying on our CR results, milk consumers in 33(68.75%) studies were exposed to moderate to high levels of carcinogenic risk through the intake of Cd and Pb, and the highest level of carcinogenic risk was reported in Pakistan, Egypt, Slovakia, Brazil, and Turkey. The high exposure to heavy metals' contamination especially Pb and Cd is a certain public health concern, therefore, maximum control measures and stricter regulation must be adopted for chemical contaminants in the dairy industry.

The presence of heavy metals contaminants in the pasteurized and sterilized milk show that they are currently used illegally in both animal husbandry and agriculture. Likewise, pasteurization and sterilization processes are not efficient for the elimination or degradation of the heavy metals addressed. In this sense, strict food safety laws should be made mandatory at every stage of milk processing units and handling to regulate and eliminate the prevalence of heavy metals under permissible limits. Further studies are required to develop safe milk processing and handling methods for the sustainable reduction of these toxic metals in milk and its products, particularly in Pakistan, Egypt, Slovakia, Brazil, and Turkey, are crucial.

Reviewer #1: We would like to thank you for the thoughtful comments and constructive suggestions, which help to improve the quality of this manuscript. We tried to answer to all valuable comments/suggestions/queries and all answers to comments are written within the document by using Red colour. 

The article is very neat and orderly, and sufficient and accurate sources have been selected. Thank you so much for your positive feedback

Do not use we Example: Line 153, “we used,” Line 156,” we included”, Line 320, “we retrieved,” Line 154“In this review we” should be omitted. It was corrected in all part of manuscript 

Tables must be modified. It should have three main lines and numbers. It was corrected 

Symbols should be used instead of color numbers and explained at the end of the table. It was corrected. 

Why didn't you do a meta-analysis? It can help the decision-making power to solve this problem and provide better results. 

 We analysied our data based on measearing health risk (carcinogenic and non-carcionogenic risk) via EPA model (Table1). Given the high heterogeneity in the data and low or moderete quality of studies (more than 70%), results were not pooled in the meta-analysis and only were interpreted based on the systematic review. Meta- analysis is not approprite when data are too heterogeneous, or bias is present in studies because the meta-analytical results coud be obscured and meaningless.In fact, it is a limitation our work. 

The implication of this review is to provide a comprehensive report on the occurrence, concentration, and potential health risk of selected heavy metals in pasteurized and sterilized milk recorded across the world in order to compare metal concentration and risk with the permissible limits. So, the conclusion was rewritten as follows:

This review is the first study that was conducted on the concentration and potential health risks of heavy metals in pasteurized and sterilized milk recorded across the world. Based on the result of this review, Cu, Cd, Zn, and Pb were the most common heavy metals, which exceeded the maximum permissible criteria in 94%, 67%, 62%, and 46% of the milk samples analyzed, respectively followed by Al, Fe, and Ni, which exceeded the local maximum permissible criteria in 30%, 16%, and 12.5% of the milk tested, respectively. Arsenic, Co, and Hg showed the lowest contamination levels and were lower than the maximum permissible limits recommended by local and international standards. Since THQ results for Pb, Hg, Co, and Cd were higher than one in milk samples examined in Slovakia, Brazil, Turkey, and Egypt, milk consumers in these countries would be exposed to non-carcinogenic risk through the continuous use of pasteurized and sterilized milk. Relying on our CR results, milk consumers in 33(68.75%) studies were exposed to moderate to high levels of carcinogenic risk through the intake of Cd and Pb, and the highest level of carcinogenic risk was reported in Pakistan, Egypt, Slovakia, Brazil, and Turkey. The high exposure to heavy metals' contamination especially Pb and Cd is a certain public health concern, therefore, maximum control measures and stricter regulation must be adopted for chemical contaminants in the dairy industry.

The presence of heavy metals contaminants in the pasteurized and sterilized milk show that they are currently used illegally in both animal husbandry and agriculture. Likewise, pasteurization and sterilization processes are not efficient for the elimination or degradation of the heavy metals addressed. In this sense, strict food safety laws should be made mandatory at every stage of milk processing units and handling to regulate and eliminate the prevalence of heavy metals under permissible limits. Further studies are required to develop safe milk processing and handling methods for the sustainable reduction of these toxic metals in milk and its products, particularly in Pakistan, Egypt, Slovakia, Brazil, and Turkey, are crucial.

The following article is one of the examples of Iran that is recommended to be used, which was not seen in your sources.

A systematic review and meta-analysis of lead and cadmium concentrations in cow milk in Iran and human health risk assessment. Environmental Science and Pollution Research. 2020 Apr;27:10147-59.

 The main purpose of this review study was to investigate Heavy metals contamination in the pasteurized and sterilized milk. 

In the study titled “A systematic review and meta-analysis of lead and cadmium concentrations in cow milk in Iran and human health risk assessment” 17 articels have been examined, among which only 3 articles have been studied on the pasteurized and sterilized milk. Those 3 studies are used in our study (Table 2). However, this study is used in interpretations in the discussion section.

Reviewer #2: Minor revision We would like to thank you for the thoughtful comments and constructive suggestions, which help to improve the quality of this manuscript. We tried to answer to all valuable comments/suggestions/queries and all answers to comments are written within the document by Italic format.

General comment: General comment: An important systematic review is presented on the presence of the main heavy metals in pasteurized and sterilized milk in the world, indicating that the estimated risk values range from moderate to high depending on the regions and conditions of the process and the system. of milk production and its productive conditions. It would be good if you included a text about the limitations of the study. Based on this valuble comment the limitation of the study was rewritten as follow: 

Although attempts were made to conduct comprehensive search strategies to include all published articles on heavy metal contamination in milk, some surveys may be lost unintentionally. Some findings like dissertations, books, and conference proceedings were not included in this study because of the low quality of results adopted in this systematic review. Given the high heterogeneity in the data, results were not pooled in the meta-analysis and only were interpreted based on the systematic review. The meta-analytical results could be obscured and meaningless, when data are too heterogeneous, or bias is present in studies. Further, the results of this study could not lead to actual conclusions about sources of heavy metal concentration in the pasteurized and sterilized milk because 77.1% of studies did not report possible sources of heavy metal.

In Abstract: in objective, indicate the years selected in the review, from that year until September 2023.

In Abstract: In the conclusion it would be important to list the first three countries or places in the world with the greatest health risk due to the consumption of pasteurized and sterilized milk. Based on this comment all relevent information was added into abstract as follow: 

Objective: Although milk and dairy products are almost complete food, they can contain toxic heavy elements with potential hazards for consumers. This review aims to provide a comprehensive report on the occurrence, concentration, and health risks of selected heavy metals in pasteurized and sterilized milk recorded worldwide.

Methods: The Preferred Reporting Items for Systematic Reviews and Meta-Analysis (PRISMA) was used to develop this systematic review. Databases included the Web of Knowledge, Scopus, Scientific Information Database, Google Scholar, and PubMed from inception until January 2023. Keywords related to the terms “Heavy metals”, “Arsenic” and “Pasteurized and sterilized milk” and “Risk Assessment” were used. The potential health risks to human health from milk daily consumption were estimated using extracted data on heavy metals concentration based on metal estimated daily intake, target hazard quotient, and carcinogenic risk. 

Results: A total of 48 potentially relevant articles with data on 981 milk samples were included in the systematic review. Atomic Absorption Spectroscopy, atomic absorption spectroscopy, inductively coupled plasma-mass spectrometry, and inductively coupled plasma-optical emission spectrometry were the most common valid methods to measure heavy metals in milk samples. Following the initial evaluation, Cu, Cd, Zn, and Pb were the most contaminants, which exceeded the maximum permissible criteria in 94%, 67%, 62%, and 46% of the milk samples tested. Relying on target hazard quotient and carcinogenic risk results, milk consumers in 33(68.75%) and 7 (14.5%) studies were exposed to moderate to high levels of carcinogenic and non-carcinogenic risk, respectively. The highest level of risk is due to the consumption of pasteurized and sterilized milk detected in Pakistan, Brazil, Egypt, Slovakia, and Turkey.

Introduction: First paragraph, it is better to consider the term food safety instead of food security, because food security in terms of public nutrition is linked to the availability, access and use of food, in this way we avoid confusion among readers. It was corrected based on this comment as follow: 

Food safety is an important challenge to maintain people's health for disease control and prevent food contamination and causing food intolerance and food poisoning (1, 2). As defined by the World Health Organization (WHO) and Food and Agriculture Organization (FAO), “food safety is a science-based discipline, process or action that prevents food from containing substances that could harm a person’s health”(2).

The nutritional importance of milk is indicated (essential fatty acids, amino acids and vitamins, etc.). I also suggest summarizing its importance from the point of view of health actions. There is important information on its protective action against coronary heart disease, stroke, diabetes and hypertension, bone health, obesity, osteoporosis, decreased risk of colorectal and bladder cancer (https://doi.org/10.3390/toxics11100809; https:/ / It was corrected based on this comment as follow:

Consuming at least three of dairy products per day has a beneficial impact on energy and nutrient intakes as well as of vitamin D, magnesium, and calcium, compared with intakes of people who consumed fewer servings of dairy products per day (6). Unsafe food containing harmful bacteria (Salmonella, Vibrio cholerae, enterohaemorrhagic Escherichia coli, and Campylobacter), viruses (Hepatitis A), parasites (tapeworms, Ascaris, Cryptosporidium, Giardia, and Entamoeba histolytica) or chemical substances (Persistent organic pollutants, heavy metals, mycotoxins, and radioactive nucleotides) causes more than 200 diseases such as diabetes, respiratory problems, hypertension, coronary heart disease, stroke, and colorectal and bladder cancer (7-9). It also promotes the immune system, good bone health, and the prevention of dental caries(8, 10).

The last line of item 2.1 is incomplete, please complete the idea. Corrected 

n 3.3.2: There are works that indicate high levels of Cd due to the use of phosphorus fertilizers in pastures and passing into milk and exceeding permissible limits: (https://doi.org/10.3390/toxics11100809;
https://doi.org/10.3390/toxics11100809;
https://doi.org/10.3390/toxics11100809;

In 3.3.3: If you see fit, you could report the As levels reported in the articles indicated for 3.3.2

 All relevent information was added as follow: 

This result can be due to uncontrolled and rapid industrial development in these countries cause elevated level of Cd in food stuffs (81). It was also evidenced that Cd generally comes from phosphate fertilizers that contain up to 53.2 mg/kg of Cd (9). Besides, equipment used in the production and packaging process is the likely other source of contamination(8, 10). The contamination of containers and equipment used in milking and pasteurization processes and environmental contamination were accounted to be among the factors affecting the elevated Cd concentration in the collected milk samples in their study (8, 10).

Although the number studies related to measuring As in pasteurized and sterilized milk across the world was so limited; several studies evaluated As concentration in raw cow’s milk which report higher values than ours from Córdoba, Argentina with mean concentration between 0.0003 and 0.0105 mg/kg (1); in Iran, in the range of 0.015 to 0.026 mg/kg(2), in Alabria, Italy, reporting an average As content of 0.038 mg/kg of raw milk, in Slovakia, with mean concentration of As < 0.03 mg/kg (1, 3). All these low concentrations of As in raw cow’s milk and pasteurized and sterilized milk are indicative that As use in the transformation of dairy products is safe for consumer and it does not pose health risks for human consumption.

In 3.4.2 standardize acronyms, either use TQT or just HQ. It was corrected

---

## [Decision Letter · Decision Letter 1]

18 Dec 2023

Concentration of Heavy Metals in Pasteurized and sterilized Milk and Health Risk Assessment across the Globe: A Systematic Review

PONE-D-23-30074R1

Dear Dr. Tavakoly Sany,

We’re pleased to inform you that your manuscript has been judged scientifically suitable for publication and will be formally accepted for publication once it meets all outstanding technical requirements.

Kind regards,

Timothy Omara, PhD

Academic Editor

PLOS ONE

Additional Editor Comments (optional):

Reviewers' comments:

Reviewer's Responses to Questions

**Comments to the Author**

1. If the authors have adequately addressed your comments raised in a previous round of review and you feel that this manuscript is now acceptable for publication, you may indicate that here to bypass the “Comments to the Author” section, enter your conflict of interest statement in the “Confidential to Editor” section, and submit your "Accept" recommendation.

Reviewer #1: All comments have been addressed

Reviewer #2: All comments have been addressed

2. Is the manuscript technically sound, and do the data support the conclusions?

Reviewer #1: Yes

Reviewer #2: Yes

3. Has the statistical analysis been performed appropriately and rigorously? 

Reviewer #1: Yes

Reviewer #2: Yes

4. Have the authors made all data underlying the findings in their manuscript fully available?

Reviewer #1: Yes

Reviewer #2: Yes

5. Is the manuscript presented in an intelligible fashion and written in standard English?

Reviewer #1: Yes

Reviewer #2: Yes

6. Review Comments to the Author

Reviewer #1: All comments have been addressed by authors and the manuscript can be published in the present form.

Reviewer #2: As the authors have substantially improved their manuscript and the review on the concentration of heavy metals in pasteurized and sterilized milk and assessment of health risks worldwide: a systematic review, I suggest being published in a journal

7. PLOS authors have the option to publish the peer review history of their article (what does this mean?). If published, this will include your full peer review and any attached files.

Reviewer #1: No

Reviewer #2: **Yes: **Jorge Castro-Bedriñana. Nutritional food security research center. Universidad Nacional del Centro del Perú

---

## [Editor Report · Acceptance letter]

26 Jan 2024

PONE-D-23-30074R1 

PLOS ONE

Dear Dr. Tavakoly Sany, 

I'm pleased to inform you that your manuscript has been deemed suitable for publication in PLOS ONE. Congratulations! Your manuscript is now being handed over to our production team.

Kind regards, 

on behalf of

Dr. Timothy Omara 

Academic Editor

PLOS ONE